# Visualizing the enzyme mechanism of mevalonate diphosphate decarboxylase

Chun-Liang Chen [1], Lake N. Paul[2,3], James C. Mermoud[1], Calvin Nicklaus Steussy[1] & Cynthia V. Stauffacher [1,4✉]

Mevalonate diphosphate decarboxylases (MDDs) catalyze the ATP-dependent-$Mg^{2+}$-decarboxylation of mevalonate-5-diphosphate (MVAPP) to produce isopentenyl diphosphate (IPP), which is essential in both eukaryotes and prokaryotes for polyisoprenoid synthesis. The substrates, MVAPP and ATP, have been shown to bind sequentially to MDD. Here we report crystals in which the enzyme remains active, allowing the visualization of conformational changes in *Enterococcus faecalis* MDD that describe sequential steps in an induced fit enzymatic reaction. Initial binding of MVAPP modulates the ATP binding pocket with a large loop movement. Upon ATP binding, a phosphate binding loop bends over the active site to recognize ATP and bring the molecules to their catalytically favored configuration. Positioned substrates then can chelate two $Mg^{2+}$ ions for the two steps of the reaction. Closure of the active site entrance brings a conserved lysine to trigger dissociative phosphoryl transfer of γ-phosphate from ATP to MVAPP, followed by the production of IPP.

[1] Department of Biological Sciences, Purdue University, West Lafayette, IN 47907, USA. [2] BioAnalysis, LLC, 1135 Dunton Street, Unit 2, Philadelphia, PA 19123, USA. [3] Biophysical Analysis Laboratory, Bindley Bioscience Center, Purdue University, West Lafayette, IN 47906, USA. [4] Purdue University Center for Cancer Research (PUCCR), Purdue University, West Lafayette, IN 47907, USA. ✉email: cstauffa@purdue.edu

Living cells produce diverse isoprenoids for maintaining cell integrity. In humans, one example is the synthesis of cholesterol by enzymes in the isoprenoid pathway, as well as to produce essential molecules such as hormones and bile acids. Similarly, microbes produce isoprenoids involved in the respiratory chain and cell wall synthesis[1,2]. Isoprenoids are synthesized from two basic building blocks, dimethylallyl diphosphate (DMAPP) and isopentenyl diphosphate (IPP). These two basic units are the ultimate products from two highly divergent isoprenoid pathways, the methylerythritol 4-phosphate (MEP) pathway found primarily in bacteria and the mevalonate (MVA) pathway found primarily in eukaryotes, although plants typically have both.

The genome sequence explosion revealed an unexpected exception to this rule. In some low GC content Gram(+) cocci, such as staphylococci, streptococci and enterococci, the mevalonate pathway exists and is essential for IPP production and pivotal for bacterial growth[3]. Thus, enzymes in the mevalonate pathway have been suggested as therapeutic targets for treatment of infectious diseases caused by these Gram(+) bacterial pathogens, especially for those bearing multidrug-resistance genes such as methicillin-resistant *Staphylococcus aureus* (MRSA) and vancomycin-resistant enterococci (VRE)[4–6], which cause a range of clinical infections[4–8]. Until now, there have only been a few therapeutic options available for treatment of such bacterial infections and resistance to these treatments is rising[9–14]. Development of antimicrobial agents has therefore become an urgent issue.

In the mevalonate pathway, the mevalonate diphosphate decarboxylase (MDD, EC: 4.1.1.33) is a rate-limiting enzyme[15], suggesting that inhibition of MDD could eliminate the products of the mevalonate pathway and accordingly shut down bacterial growth. In addition, an in vitro study had shown feedback regulation by mevalonate-5-diphosphate (MVAPP) of the mevalonate kinase (MK) from *Streptococcus pneumonia*[16]. This implies that accumulation of MVAPP caused by inhibition of MDD can down-regulate the upstream MK enzymes, also indicating that the mevalonate pathway can be effectively inhibited by targeting the MDD enzyme. A broad-spectrum substrate-mimicking inhibitor

of MDD, 6-fluoromevalonate diphosphate (FMVAPP), has been identified. This inhibitor, however, binds to the highly conserved MVAPP-binding site in the MDD family of proteins (MDDs)[17–22], including human MDD. To avoid side effects on humans when treating bacterial infections, structure-based drug development toward an MDD with both sensitivity and specificity is needed[23].

MDDs perform a sequential ordered bi-substrate mechanism with MVAPP as the first substrate[17,24,25] followed by ATP binding. The γ-phosphate group of ATP is transferred to MVAPP to make the 3′-phosphate-MVAPP intermediate, which is then subjected to dephosphorylation and decarboxylation to produce IPP. Although it was proposed that the ATP-dependent decarboxylation of MVAPP[26] might be initiated via deprotonation of 3′-OH of MVAPP by a conserved aspartic acid (D282 in MDD from *Enterococcus faecalis*, and D283 in MDD from *Staphylococcus epidermidis*), followed by the transfer of the γ-phosphate of ATP to MVAPP[27], recent mutagenesis studies on MDD from *Sulfolobus solfataricus* (MDD$_{SS}$) have demonstrated that the catalytic Asp residue may function in dephosphorylation and decarboxylation of the 3′-phosphate-MVAPP intermediate, rather than deprotonation of the 3′-OH group of MVAPP[28]. However, the phosphoryl transfer efficiencies of the D281T or D281V mutants of MDD$_{SS}$ are lower than the wild-type MDD$_{SS}$, implying some involvement of the Asp in this step.

In addition, although the reaction requires magnesium ions for catalysis[17,18,29], no metal ions have been found in previously published MDD structures[17,18]. This indicates that current structural models of MDDs may not provide sufficient information for elucidating the enzyme mechanism or designing effective inhibitors of MDDs.

In this study, we utilized MDD from *Enterococcus faecalis* V583 (MDD$_{EF}$), a VRE strain, and investigated the enzyme from structural, functional and biophysical points of view. Our present results suggest that the substrate-binding mechanism of MDDs involves programmed conformational rearrangements in and around the active site, including movements of the β10-α4 loop, the phosphate-binding loop, and three helices, α1, α2, and α4. In the analysis of these conformational changes, the critical role of a conserved lysine (K187) in catalysis also became clear. These findings provide insight into a detailed MDD enzyme mechanism, which then can shed light on specific drug development against the MDD proteins of drug-resistant pathogenic bacteria[5,7,9,10,22,30,31].

## Results

**Analysis of MDD$_{EF}$ crystal structures in reaction steps**. The investigation of the structural details of the MDD$_{EF}$ mechanism was aided by the discovery of crystallization conditions under which the introduction of substrates and cofactors by soaking resulted in activity in the crystal. These MDD$_{EF}$ crystals were grown at a high ammonium sulfate concentration (1.6 M)[19] but subjected to buffer exchange into PEG3350 solutions before soaking with ligands. In this study, crystal structures of apo (MDD$_{EF}$-SO$_4^{2-}$), MVAPP-bound (MDD$_{EF}$-MVAPP), open (MDD$_{EF}$-MVAPP-AMPPCP-Mg$^{2+}$), and closed conformations of MDD$_{EF}$ (MDD$_{EF}$-MVAPP-ADPBeF$_3$-Mg$^{2+}$ and MDD$_{EF}$-MVAPP-ADP-SO$_4^{2-}$-Co$^{2+}$) were obtained by providing the apo enzyme crystals with the appropriate sets of substrates. A nearly full-length (residue 1–326) structural model was generated from the crystal structure of MDD$_{EF}$-MVAPP-ADP-SO$_4^{2-}$-Co$^{2+}$, showing that MDD$_{EF}$ folds into a two-layer sandwich architecture with α and β secondary structure elements (α β class) (Fig. 1), similar to those for published MDD structures. Information

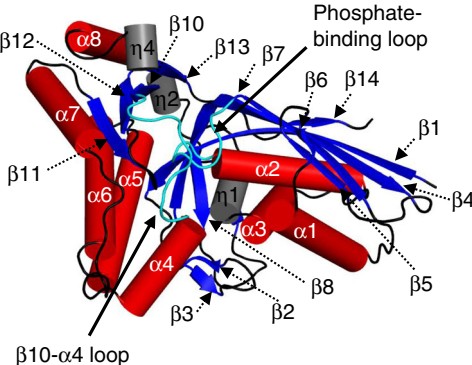

**Fig. 1 MDD$_{EF}$ structure.** The crystal structure of MDD$_{EF}$ in complex with MVAPP, ADPBeF$_3$, and Mg$^{2+}$ (MDD$_{EF}$-MVAPP-ADPBeF$_3$-Mg$^{2+}$) was analyzed by PROMOTIF[61]. Secondary structure elements, α-helices (red cylinders), 3$_{10}$-helices (gray cylinders) and β-sheets (blue arrows) are drawn and labeled as α, η, and β. The β10-α4 loop and the phosphate-binding loop are shown in cyan. The range of amino acid numbers of each secondary structural element is listed as following: β1(1–15), β2(20–22), β3 (27–28), β4(32–49), β5(56–59), β6(62–63), β7(89–96), β8(149–153), β9 (162–165), β10(175–180), β11(277–280), β12(287–292), β13(312–315), β14 (322–324), α1(66–81), α2(105–120), α3(127–137), α4(191–201), α5 (205–224), α6(228–246), α7(258–272), α8(298–303), η1(139–142), η2 (171–174), η3(293–297), η4(308–311).

**Table 1 Data collection and refinement statistics.**

| | $MDD_{EF}$-$SO_4^{2-}$ | $MDD_{EF}$-MVAPP | $MDD_{EF}$-MVAPP-AMPPCP-$Mg^{2+}$ | $MDD_{EF}$-MVAPP-ADPBeF$_3$-$Mg^{2+}$ | $MDD_{EF}$-MVAPP-ADP-$SO_4^{2-}$-$Co^{2+}$ | $MDD_{EF}$-MVAPP-ADP-$SO_4^{2-}$-$Co^{2+}$ |
|---|---|---|---|---|---|---|
| **Data collection** | | | | | | |
| Space group | $P2_12_12$ | $P2_12_12$ | $P2_12_12$ | $P2_12_12$ | $P2_12_12$ | $P2_12_12$ |
| Cell dimensions | | | | | | |
| $a, b, c$ (Å) | 82.4, 97.9, 45.7 | 79.3, 97.4, 45.8 | 80.0, 97.0, 46.0 | 79.6, 98.6, 45.6 | 80.0, 98.1, 45.9 | 79.9, 98.0, 45.9 |
| $\alpha, \beta, \gamma$ (°) | 90, 90, 90 | 90, 90, 90 | 90, 90, 90 | 90, 90, 90 | 90, 90, 90 | 90, 90, 90 |
| Resolution (Å)[a] | 50–1.8 (1.86–1.80) | 50–1.7 (1.76–1.70) | 30–2.1 (2.12–2.05) | 30–2.1 (2.18–2.10) | 50–2.0 (2.02–1.95) | 30–2.4 (2.43–2.35) |
| $R_{merge}$ | 6.1 (50.1) | 5.0 (43.8) | 4.1 (51.3) | 5.8 (52.3) | 5.4 (48.9) | 7.7 (49.5) |
| $I/\sigma I$ | 32.3 (3.1) | 40.3 (4.1) | 38.6 (2.5) | 23.3 (2.2) | 37.8 (3.5) | 40.0 (4.5) |
| $CC_{1/2}$ | 0.980 (0.926) | 0.982 (0.931) | 0.964 (0.846) | 0.954 (0.832) | 0.980 (0.917) | 0.984 (0.920) |
| Completeness (%) | 99.2 (100) | 99.5 (100) | 96.6 (98.5) | 99.8 (99.8) | 99.2 (100) | 99.5 (99.9) |
| Redundancy | 7.0 (6.9) | 6.8 (6.8) | 5.1 (4.9) | 4.5 (4.0) | 6.8 (6.7) | 12.7 (12.3) |
| **Refinement** | | | | | | |
| Resolution (Å) | 30–1.8 | 30–1.7 | 30–2.0 | 30–2.1 | 34–2.0 | 27–2.3 |
| No. of reflections | 34,579 | 39,789 | 22,148 | 21,068 | 26,418 | 15,594 |
| $R_{work}/R_{free}$ | 0.186/0.211 | 0.153/0.175 | 0.193/0.217 | 0.173/0.192 | 0.196/0.219 | 0.183/0.222 |
| No. of atoms | | | | | | |
| Protein | 2607 | 2487 | 2528 | 2528 | 2528 | 2528 |
| Ligand/ion | 10 | 18 | 50 | 51 | 52 | 52 |
| Water | 234 | 382 | 91 | 146 | 187 | 187 |
| $B$-factors | | | | | | |
| Protein | 23.3 | 15.7 | 31.1 | 27.4 | 27.6 | 34.5 |
| Ligand/ion | 28.6 | 10.2 | 34.1 | 22.0 | 20.8 | 30.9 |
| Water | 31.5 | 27.4 | 30.9 | 29.6 | 29.8 | 35.8 |
| R.m.s. deviations | | | | | | |
| Bond lengths (Å) | 0.005 | 0.006 | 0.01 | 0.012 | 0.01 | 0.01 |
| Bond angles (°) | 0.78 | 1.02 | 1.02 | 1.13 | 1.00 | 1.00 |
| PDB code | 6E2S | 6E2T | 6E2U[b] | 6E2V | 6E2W | 6E2Y[b] |

[a]Values in parentheses indicate the highest-resolution shell.
[b]Data were collected on the home source with CuKα X-ray radiation.

regarding data collection and refinement statistics is summarized in Table 1.

The apo form of $MDD_{EF}$ ($MDD_{EF}$-$SO_4^{2-}$) has almost identical secondary structure elements compared with other MDDs. However, the phosphate-binding loop (97–104) and the β10-α4 loop (183–190) cannot be determined in this apo structure, which is different from other published apo-MDD structures[17,19,32] (Supplementary Table 1). Compared with them, no crystal-packing contacts were found in these two loop regions of the apo-$MDD_{EF}$ presented here, implying that these two loops may be dynamic or disordered in the structure. In the $MDD_{EF}$-MVAPP structure, the conformation of the β10-α4 loop (Fig. 2a) and MVAPP can now be clearly seen (Fig. 2b). Although the binding configuration of MVAPP and the β10-α4 loop are similar to what had been determined in other published MVAPP-bound MDD structures[18], the β10-α4 loop appears to be stabilized by helix α4, which interacts with MVAPP through the bonds between the pyrophosphate of MVAPP and the residues S191 and R192 of helix α4. These interactions may confine the preceding β10-α4 loop to a certain configuration and accomplish the first step in the reaction.

To investigate any structural rearrangements upon ATP binding in the next reaction step, $MDD_{EF}$-MVAPP crystals were soaked with ATP analogs (AMPCP or ADPBeF$_3$) to mimic two substrate-bound $MDD_{EF}$ structures before ATP cleavage and in the transition step. This resulted in two different conformations— open ($MDD_{EF}$-MVAPP-AMPPCP-$Mg^{2+}$) and closed ($MDD_{EF}$-MVAPP-ADPBeF$_3$-$Mg^{2+}$) conformations of $MDD_{EF}$ (Fig. 2c–f). In the open conformation, the β10-α4 loop and the phosphate-binding loop are now ordered in the active site (Fig. 2c), as are the

ligands, MVAPP and ATP/AMPPCP (Fig. 2d). This suggested that the MgATP binding can trigger conformational changes of the phosphate-binding loop to take up a position poised over the active site but still open. This configuration was also observed in our previously published $MDD_{EF}$-ATP structure (5V2L)[24]. Note that these two loops do not directly contact active site ligands in the open $MDD_{EF}$ conformation, suggesting that substrate binding may reshape the general active site environment and stabilize helices α2 and α4, which leads to overall stabilization of the phosphate-binding loop and the β10-α4 loop, respectively. Interestingly, an extra density not explained by MVAPP or AMPPCP appears to be between the phosphate groups of substrates. This density is surrounded by five oxygen atoms (one from MVAPP, three from AMPPCP and one from γ-OH of S106), implicating that the extra electron density is a magnesium ion (Fig. 2d).

In the closed conformation of $MDD_{EF}$ ($MDD_{EF}$-MVAPP-ADPBeF$_3$-$Mg^{2+}$) (Fig. 2e, f), the β10-α4 loop and the phosphate-binding loop have become close around the active site entrance. Ligand densities in the active site were clearly distinguishable (Fig. 2f). In this structure, not one but two extra spherical densities not belonging to MVAPP or ADPBeF$_3$ were identified. One spherical density is surrounded by six oxygen atoms (one from MVAPP, two from ADPBeF$_3$, one from the γ-OH of S106 and one from water), and the other one is coordinated by five oxygen atoms (three from water and two from MVAPP) and one fluorine atom from BeF$_3^-$. These two spherical densities are suggested to be the magnesium ions required for the MDD reaction (Fig. 2f). Two metal-binding sites were later confirmed with a distinct anomalous signal of cobalt ions in the complex

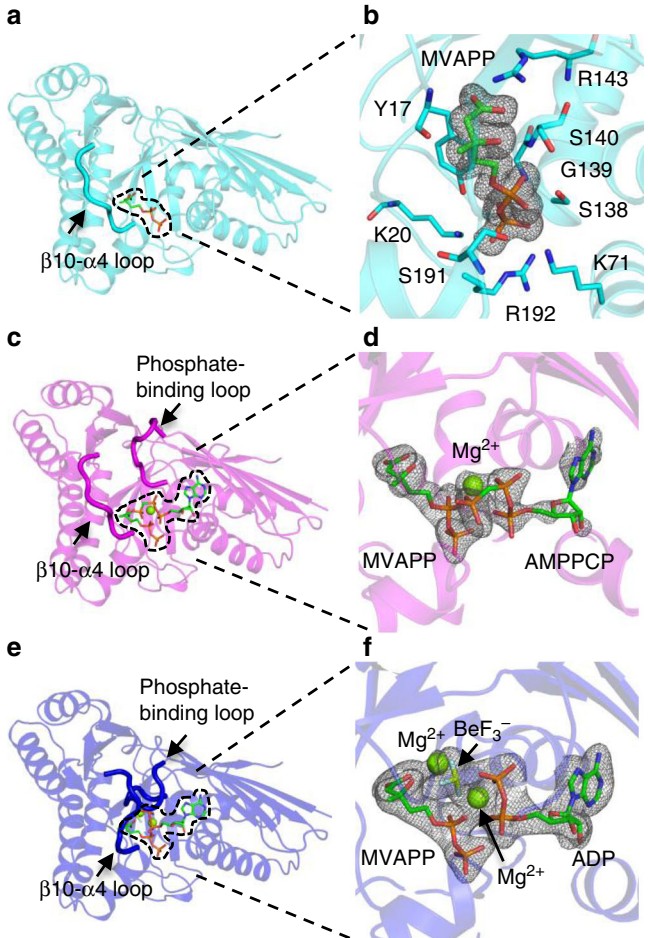

**Fig. 2 Structural models of bound forms of MDD$_{EF}$. a** Structural model of MDD$_{EF}$-MVAPP. The β10-α4 loop is indicated by a black arrow. **b** SA-ligand-omit map of MVAPP of MDD$_{EF}$-MVAPP. The map is presented at a sigma level of three (3σ). MVAPP and the interacting residues in the active site of MDD$_{EF}$ are shown as stick models. **c** Structural model of MDD$_{EF}$-MVAPP-AMPPCP-Mg$^{2+}$. The β10-α4 loop and the phosphate-binding loop are indicated by black arrows. **d** SA-omit map (3σ) of ligands of MDD$_{EF}$-MVAPP-AMPPCP-Mg$^{2+}$. **e** Structural model of the MDD$_{EF}$-MVAPP-ADPBeF$_3$-Mg$^{2+}$. The β10-α4 loop and the phosphate-binding loop are indicated by black arrows. **f** SA-omit map (3σ) of ligands in MDD$_{EF}$-MVAPP-ADPBeF$_3$-Mg$^{2+}$. Ligand-binding sites in **a**, **c**, and **e** are indicated by dashed lines.

structure of MDD$_{EF}$-MVAPP-ADP-SO$_4^{2-}$-Co$^{2+}$ (described in the next section).

ADPBeF$_3$ is an ATP analog in which BeF$_3^-$ mimics the γ-phosphoryl group of ATP[33]. In MDD$_{EF}$-MVAPP-ADPBeF$_3$-Mg$^{2+}$, BeF$_3^-$ was located between the Oβ atom of ADP and 3′-oxygen of MVAPP, and the angle of Oβ-Be-3′-O was determined to be nearly 180° (178°), suggesting that the BeF$_3$ molecule is located in an in-line phosphoryl transfer position. The distance of the 3′-oxygen of MVAPP to the beryllium atom was determined to be 3.7 Å, within a possible phosphate transfer distance (Supplementary Fig. 1). The distance of Oβ to beryllium is 2.0 Å (Supplementary Fig. 1), which is longer than a normal P–O single bond of 1.6 Å, suggesting that ADPBeF$_3$ in our closed complex structure of MDD$_{EF}$-MVAPP-ADPBeF$_3$-Mg$^{2+}$ is situated in a pre-phosphoryl transfer state. The closed MDD$_{EF}$ structure represents the state just as the enzymatic transfer begins, whereas the complex structure of MDD$_{EF}$-MVAPP-AMPPCP-Mg$^{2+}$ might represent an earlier state upon ATP binding.

The closed conformation of MDD$_{EF}$ may restrict bulk water diffusing into the active site to prevent wasteful phosphoryl transfer[34] during the MDD enzymatic reaction, whereas the active site in MDD$_{EF}$-MVAPP-AMPPCP-Mg$^{2+}$ is still accessible to solvent. In addition, the distance between 3′-oxygen of MVAPP and Pγ of AMPPCP (6.1 Å) is substantially longer than the distance between 3′-oxygen of MVAPP and Be of ADPBeF$_3$ in MDD$_{EF}$-MVAPP-ADPBeF$_3$-Mg$^{2+}$ (3.7 Å). These differences suggest that phosphoryl transfer is unlikely to take place in the open conformation of MDD$_{EF}$ as shown in MDD$_{EF}$-MVAPP-AMPPCP-Mg$^{2+}$. The occupancy of AMPPCP in the open MDD$_{EF}$ structure was about 70% of that of the closed structure where the occupancy of ADPBeF$_3$ is 100% and AMPPCP showed fewer contacts with MDD$_{EF}$. A large $K_{IAMPPCP}$ value (>1 mM) determined from the previous kinetic studies on chicken MDD also supports these observations[25]. This implies that the binding affinity of ATP to the open conformation of MDD$_{EF}$ may be initially weak and the active site has to be rearranged for accommodation of this substrate.

**Two metal-binding sites in the closed MDD$_{EF}$-ligand structure.** In the closed conformation of MDD$_{EF}$-MVAPP-ADPBeF$_3$-Mg$^{2+}$, the two extra spherical densities observed were proposed to be two magnesium ions, using the evidence of the coordination by surrounding atoms. However, magnesium does not produce a detectable anomalous signal for identification and its electron density cannot be easily distinguished from a water molecule. In order to confirm whether there are two metal-binding sites formed during the MDD enzymatic reaction, we prepared ligand-bound MDD$_{EF}$ crystals under buffer conditions with cobalt substituting for magnesium. Cobalt has been confirmed as an alternative cofactor for MDD protein catalysis[35], and ordered cobalt ions in a crystal lattice can produce a detectable anomalous signal even with a home-source X-ray radiation[36]. To confirm the previous conclusion that cobalt can substitute for magnesium in the buffer condition, enzymatic experiments were conducted and the kinetic parameters of MDD$_{EF}$ were determined with cobalt ($V_{max} = 9.5 \pm 0.3$ μMol min$^{-1}$ mg$^{-1}$, $K_{mMgATP} = 188 \pm 13$ μM, $K_{mMVAPP} = 39.3 \pm 4.0$ μM) (Supplementary Fig. 2a, b and Supplementary Table 2). The $V_{max}$ value is ~70% compared with the $V_{max}$ value under conditions with magnesium, consistent with the previously published results[35]. A decrease in MDD$_{EF}$ enzymatic activity is not due to a reduced enzymatic activity of pyruvate kinase in the coupled reaction assay since the assay can detect a reaction rate of $85.4 \pm 1.6$ μMol min$^{-1}$ mg$^{-1}$ under the conditions with cobalt (Supplementary Fig. 2c).

The anomalous diffraction data for the MDD$_{EF}$ bound to MVAPP, ADP, SO$_4^{2-}$, and cobalt (MDD$_{EF}$-MVAPP-ADP-SO$_4^{2-}$-Co$^{2+}$) were collected locally with CuKα X-ray radiation; a higher-resolution diffraction data set from the same crystal was also collected for model building (Table 1 and Fig. 3a). The overall structure, the phosphate-binding loop, the β10-α4 loop (Fig. 3a) and ligands in the active site were all well determined in the high resolution map (Fig. 3b, SA-ligand-omit map at 3 σ), as were the two spherical electron densities located between MVAPP and ADP-SO$_4^{2-}$ suggested to belong to cobalt (Fig. 3b). This 2.0-Å model was then used to provide phases for the anomalous dispersion data. The resulting anomalous difference map also shows two strong and distinct cobalt positions from the anomalous signal located exactly at the two suggested metal-binding sites (Fig. 3c, anomalous difference map at 5 σ).

The crystal structures of MDD$_{EF}$-MVAPP-ADPBeF$_3$-Mg$^{2+}$ and MDD$_{EF}$-MVAPP-ADP-SO$_4^{2-}$-Co$^{2+}$ were superimposed and analyzed (Supplementary Fig. 3a, b). Although the Mg$^{2+}$ and Co$^{2+}$ ions near D282 were located at slightly different positions

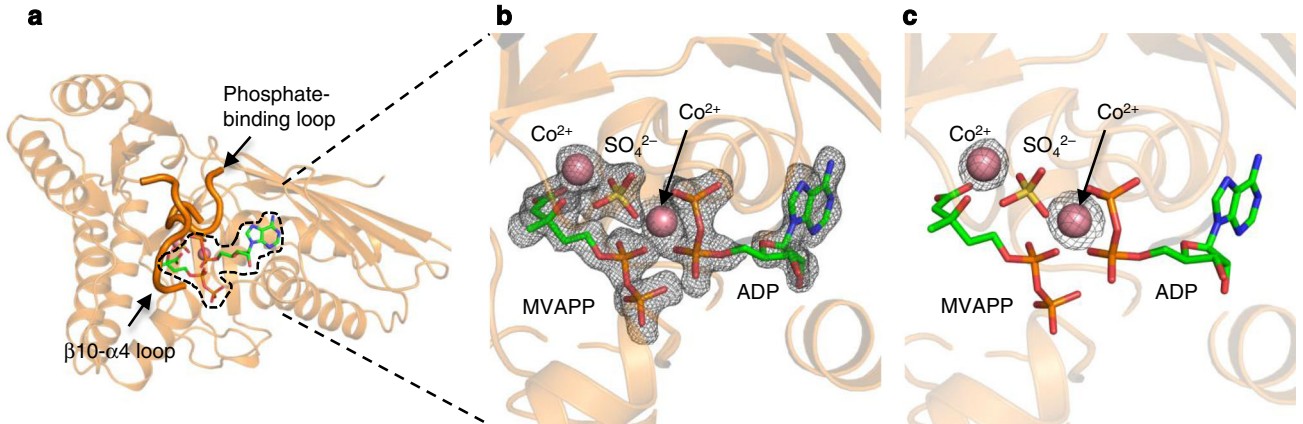

**Fig. 3 The two metal-binding sites of MDD$_{EF}$. a** Structural model of MDD$_{EF}$-MVAPP-ADP-SO$_4^{2-}$-Co$^{2+}$. The β10-α4 loop and the phosphate-binding loop in the closed configuration are indicated by black arrows. The binding site of ligands is indicated with a dashed line. **b** SA-omit map (3σ) of ligands (MVAPP, ADP, SO$_4^{2-}$ and Co$^{2+}$). **c** An anomalous difference map was derived from the home-source (CuKα) data set. The anomalous difference map is contoured at a sigma level of five (5σ).

(Supplementary Fig. 3b), this could be due to the slight dislocation of the sulfate ion that experiences steric pressure and charge repulsion from the diphosphate group of ADP. In summary, MDD$_{EF}$-MVAPP-ADP-SO$_4^{2-}$-Co$^{2+}$ has two cobalt ions in the active site, which are located at similar positions as two spherical densities in MDD$_{EF}$-MVAPP-ADPBeF$_3$-Mg$^{2+}$, suggesting that two magnesium ions are chelated in the active site during the enzymatic reaction. A signal for the metals was only observed in the crystal structures of MDD$_{EF}$-MVAPP-ADP-SO$_4^{2-}$-Co$^{2+}$ and MDD$_{EF}$-MVAPP-ADPBeF$_3$-Mg$^{2+}$ but not in either apo-MDD$_{EF}$ or MVAPP-bound MDD$_{EF}$ structures, suggesting that two metal-binding sites are transiently formed only in the closed conformation of the MDD$_{EF}$-ligand complex.

**Dynamic changes of MDD$_{EF}$ upon substrate binding**. Four different states have been identified in the MDD$_{EF}$ enzymatic reaction—an apo form (MDD$_{EF}$-SO$_4^{2+}$), a first substrate-bound structure (MDD$_{EF}$-MVAPP), an open complex structure mimicking a two substrate-bound intermediate (MDD$_{EF}$-MVAPP-AMPPCP-Mg$^{2+}$), and a closed complex structure just before the catalytic transition (MDD$_{EF}$-MVAPP-ADPBeF$_3$-Mg$^{2+}$). Structural analysis of these MDD$_{EF}$ crystal structures revealed dynamics of five regions upon substrate/ligand binding: the phosphate-binding loop, the β10-α4 loop, and three helices that border the active site, α1, α2, and α4.

The phosphate-binding loop and the β10-α4 loop were observable both in the ligand-bound open (MDD$_{EF}$-MVAPP-AMPPCP-Mg$^{2+}$) and closed conformations (MDD$_{EF}$-MVAPP-ADPBeF$_3$-Mg$^{2+}$) of MDD$_{EF}$ (Fig. 4a). In the open structure, these two loops do not interact with ligands, whereas in the closed conformation of MDD$_{EF}$, two loops enclose the active site and form contacts with ligands (Supplementary Table 3). A 9.4-Å distance change in the β10-α4 loops and a 10.6-Å distance change in the phosphate-binding loops (Fig. 4a) were observed between these two steps. A whole-structure analysis of Cα positions between the open and closed conformations is shown in Fig. 4b, indicating significant structural changes in these two loop regions.

Key interactions between MDD$_{EF}$ and ligands in the three ligand-bound structures are summarized in Supplementary Table 3. Residues in α1, α2, and α4 interact with ligands differently in these ligand-bound structures of MDD$_{EF}$. The gain or loss of MDD$_{EF}$-ligand interactions indicates that the three helices, α1, α2, and α4, move to accommodate substrates during

the enzymatic reaction. Figure 5a shows the positions of helix α1 (left), helix α2 (middle), and helix α4 (right). Figure 5b–d shows the individual positions of helix α1 (Fig. 5b), helix α2 (Fig. 5c), and helix α4 (Fig. 5d) in the four structures, MDD$_{EF}$-SO$_4^{2-}$ (left, representing "State I"), MDD$_{EF}$-MVAPP (middle-left, representing "State II"), MDD$_{EF}$-MVAPP-AMPPCP-Mg$^{2+}$ (middle-right, representing "State III"), and MDD$_{EF}$-MVAPP-ADPBeF$_3$-Mg$^{2+}$ (right, representing "State IV"). By superimposing these four crystal structures, movements of key residues/helical centers in helix α1 (Fig. 5d, left), helix α2 (Fig. 5d, middle), and helix α4 (Fig. 5d, right) upon substrate binding were proposed and are described below.

Q68 and K71 in helix α1 interact with MVAPP and the ATP analogs. Upon MVAPP binding, K71 interacts with the pyrophosphate group of MVAPP, which is consistent with the movement of α1 from state I to state II (Fig. 5b). The movement of α1 accordingly relocated Q68 to remodel a better ATP-binding pocket, agreeing with our previous results[24]. Although these two residues had been known to form contacts with substrates[18], the structural dynamics of MDD$_{EF}$ helix α1 upon substrate binding is now observed in these structures (Fig. 5b, state I and state II). In higher organisms (e.g., MDD from *Arabidopsis thaliana*), the Lys residue in helix α1 is replaced with Arg for the interaction with MVAPP[37], similar to R192 in MDD$_{EF}$. Such evolutionary convergence may still facilitate the movement of helix α1 of MDDs from higher organisms upon MVAPP binding.

Helix α2 moves towards the active site in an ordered manner when binding to the substrates (Fig. 5c). The movement of helix α2 upon MVAPP binding was unexpected since no direct contact was found between helix α2 and MVAPP. Helix α1 is close to helix α2, so the movement of α1 could affect α2 in order to facilitate the subsequent binding of ATP. A conserved S106 (S107 in MDD$_{SE}$) in the N terminus of helix α2 is also positioned differently in these four structures, with a change in side-chain orientation (Fig. 5c, e, middle). The role of this conserved serine was previously assigned as a key residue for the binding of the ATP γ-phosphoryl group as interpreted from the crystal structure of MDD$_{SE}$-FMVAPP-ATPγS[18] (Supplementary Fig. 4), and mutation of this residue had been shown to decrease MDD enzymatic activity drastically[17]. Surprisingly, we have found that this conserved S106 composes a part of one of the metal-binding sites in the open (MDD$_{EF}$-MVAPP-AMPPCP-Mg$^{2+}$) and closed (MDD$_{EF}$-MVAPP-ADPBeF$_3$-Mg$^{2+}$) conformations of MDD$_{EF}$,

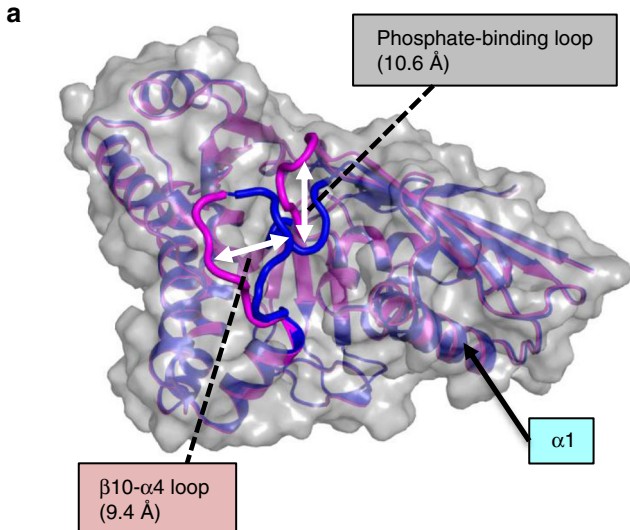

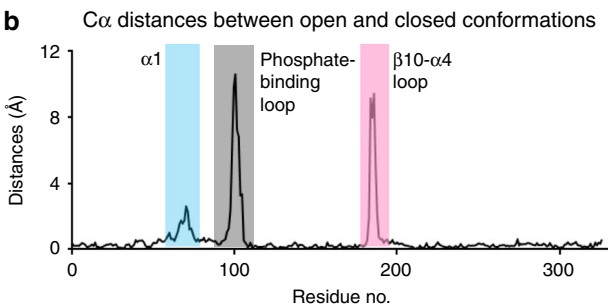

**Fig. 4 Differences between open and closed MDD$_{EF}$ bound with ligands. a** Superposition of open (MDD$_{EF}$-MVAPP-AMPPCP-Mg$^{2+}$, magenta) and closed (MDD$_{EF}$-MVAPP-ADPBeF$_3$-Mg$^{2+}$, blue) conformations of MDD$_{EF}$. A gray surface represents the envelope of the apo-MDD$_{EF}$. The greatest distance between the β10-α4 loops is 9.4 Å (left white arrow) determined by measuring the distance between K187 Cα carbons in the open and closed structures. The greatest distance between the phosphate-binding loops is 10.6 Å (right white arrow) determined by measuring the distance between A101 Cα carbons in open and closed structures. **b** Distances of Cα atoms between open and closed conformations. The structural models of open and closed MDD$_{EF}$ were Cα-aligned to apo-MDD$_{EF}$ with r.m.s. deviation values of 0.35 and 0.40 Å, respectively. The distances between corresponding Cα atoms in both structures is plotted as a function of residue number. The regions of helix α1 (α1, 66–81), the phosphate-binding loop (97–104) and the β10-α4 loop (183–190) are highlighted in cyan, gray, and pink, respectively. Source data are provided as a Source Data file.

taking the same position as previously assigned to the γ-phosphate.

Helix α4 moves from state I to state II (Fig. 5e, right), with the conserved S191 on the N terminus of helix α4 forming contacts with the pyrophosphate of MVAPP (Fig. 2b; Supplementary Table 3)[17]. Although the position of helix α4 in the three MVAPP-bound MDD$_{EF}$ structures showed no significant change (Fig. 5e, right), the side-chain orientation of S191 in the closed conformation (MDD$_{EF}$-MVAPP-ADPBeF$_3$-Mg$^{2+}$) changes to interact with conserved K187, resulting in loss of contacts with MVAPP. In summary, the angular changes in helix α1, α2, and α4 are consistent with the dynamic interactions between key residues and ligands (Supplementary Tables 4 and 5). These differences in the apo and ligand-bound MDD$_{EF}$ structures here reveal a coordinated set of programmed conformational rearrangements around the active site region upon each step of substrate binding during the catalytic reaction.

**The role of conserved K187 in the non-conserved β10-α4 loop.** In the closed MDD$_{EF}$ structure (MDD$_{EF}$-MVAPP-ADPBeF$_3$-Mg$^{2+}$), the non-conserved β10-α4 loop covers the active site (Fig. 4), and K187 in the β10-α4 loop extends its side-chain into the active site to interact with S191 and the bridging oxygen of ADPBeF$_3$ (Fig. 5d, right). Lysine/arginine residues are known for substrate binding or neutralizing the negatively charged active site in kinases[38,39]. In MDD$_{EF}$, K71 and R144 function for MVAPP binding and may also fulfill a neutralization role (Fig. 2b). The K187 residue is conserved among the MDD family of proteins (Fig. 6a), and mutation of the corresponding lysine (K208A in rat MDD) resulted in a dead enzyme[40]. This experimental evidence indicates that this lysine is critical for the enzyme reaction. To investigate the function of this lysine residue, a K187A mutant of MDD$_{EF}$ was created and the K187A mutant protein was purified and examined (see "Methods" section; Supplementary Fig. 5). The K187A mutant showed a nearly-dead enzymatic activity of 0.35% compared with the wild-type enzyme at saturated substrate concentrations. A ~300 fold decrease in enzymatic activity confirms the essentiality of K187 in the reaction (Fig. 6b). To differentiate the role of K187 in either catalysis or substrate binding, ITC was then employed to determine $K_d$ values under different conditions[41]. All the derived thermodynamic parameters are listed in Supplementary Table 6.

The $K_{dATPγS}$ values were determined under different conditions with the K187A mutant alone or the K187A mutant pre-incubated with MVAPP (see "Methods" section). The $K_{dATPγS}$ value between ATPγS and the K187A mutant is 182 ± 36 μM, similar to the value of $K_{dATPγS}$ of wild-type MDD$_{EF}$ (215 ± 8 μM)[24]. However, in the presence of MVAPP, $K_{dATPγS}$ is 58.2 ± 13.2 μM (Fig. 6c), which is only about two-fold higher than the $K_{dATPγS}$ value of wild-type MDD$_{EF}$ (25.4 ± 5.5 μM)[24], suggesting that K187 has key roles mainly in catalysis. The side-chain of K187 is at a hydrogen bond distance to the β-γ bridging oxygen in MDD$_{EF}$-MVAPP-ADPBeF$_3$-Mg$^{2+}$. This positively charged side-chain, together with the Mg$^{2+}$ ions, could relieve a negatively charged environment built up upon substrate binding and during the phosphoryl transfer reaction[42]. The present results suggest that K187 can be transiently involved in the reaction, and the conserved S191 may confer initially a site for MVAPP binding and later an anchoring point for accommodating K187 in the active site for enzyme catalysis.

**Discussion**

The four structures presented here clearly represent different states of MDD$_{EF}$ during the enzymatic reaction. Based on these experimental observations, we suggest a model for interpreting the physical steps of the MDD$_{EF}$ enzymatic reaction upon the ordered substrate binding of the sequential ordered bi-substrate mechanism. In the physical mechanism, the phosphate-binding loop and the β10-α4 loop may be initially dynamic or disordered in an apo structure (Fig. 7a, left). Upon MVAPP binding, induced structural changes take place in the β10-α4 loop and three α-helices (α1, α2, and α4) (Fig. 7a, middle-left). The movements of helix α2 and α1 reposition S106 and Q68 to facilitate ATP and metal binding, as experimentally supported by previous ITC results[24]. The initial binding of ATP stabilizes the phosphate-binding loop (Fig. 7a, middle-right). The phosphate tail of ATP, S106 and the pyrophosphate of MVAPP form the first metal-binding site. Subsequent structural rearrangement occurs and the phosphate-binding loop bends down to "drag" ATP into its catalytically favored position (Supplementary Fig. 6a, b). The recognition of the β-γ-bridging oxygen of ATP by the phosphate-binding loop may therefore be a checkpoint during enzyme catalysis. The second metal is fit into the active site, followed by the two loops closing the substrate entrance (Fig. 7a, right). In the

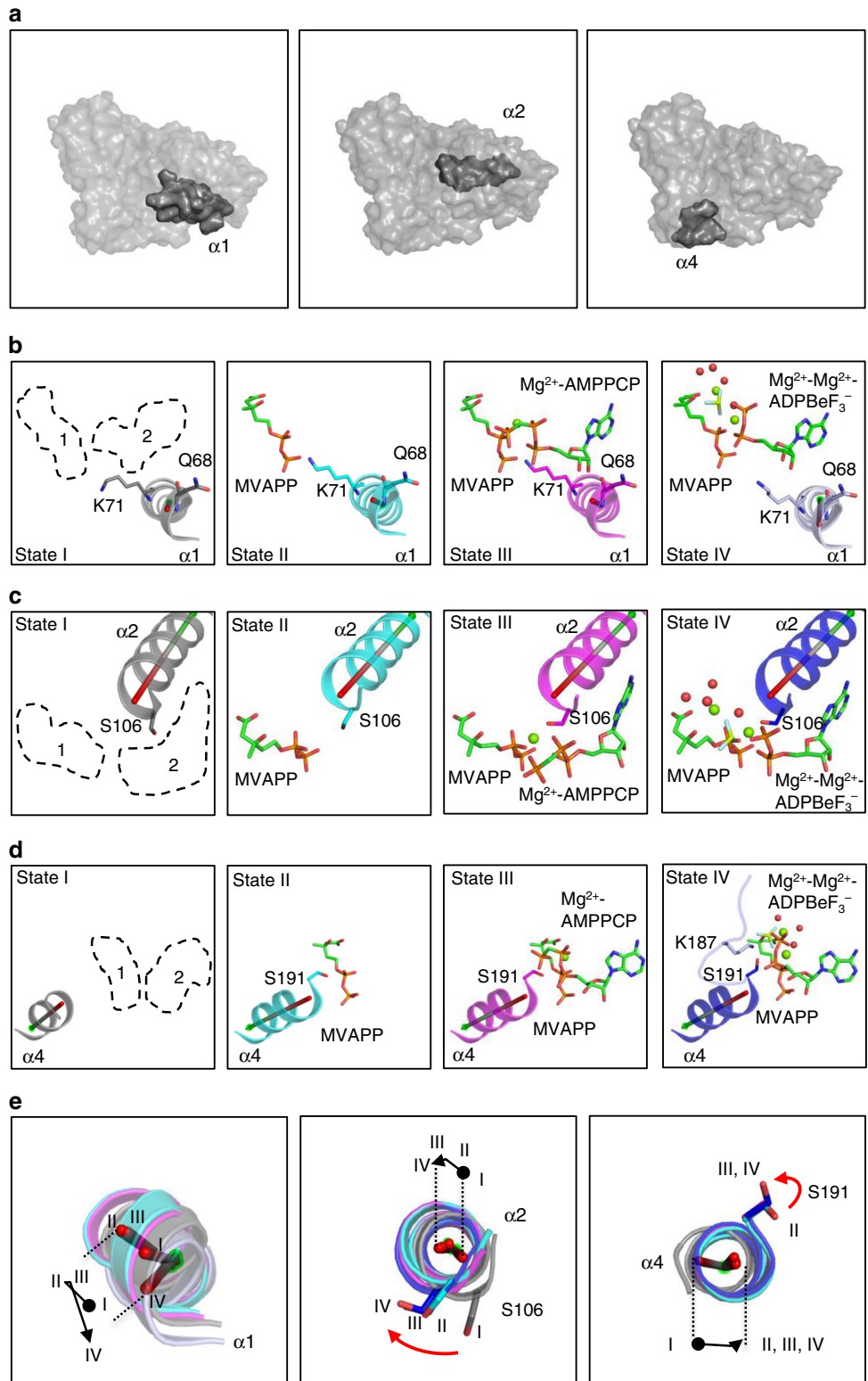

**Fig. 5 Movements of helix α1, α2, and α4 in the unbound and ligand-bound MDD$_{EF}$. a** Locations of helix α1 (left), α2 (middle), and α4 (right) in MDD$_{EF}$. **b** Positions of helix α1 and key residues (Q68 and K71) in four different structures (left: apo-MDD$_{EF}$, gray, state I; middle-left: MDD$_{EF}$-MVAPP, cyan, state II; middle-right: MDD$_{EF}$-MVAPP-AMPPCP-Mg$^{2+}$, magenta, state III; right: MDD$_{EF}$-MVAPP-ADPBeF$_3$-Mg$^{2+}$, blue, state IV). The binding sites of MVAPP (1) and ATP (2) are circled with dashed lines. The center of helix α1 is shown as a rod colored in red (N terminus) and green (C terminus). Magnesium ions and water molecules are shown as green and red spheres, respectively. **c** Position of helix α2 and key residues (S106) in four different structures as described in **b**. **d** Position of helix α4 and key residues (S191) in four different structures as described in **b**. K187 in the β10-α4 loop (right panel) is shown as a stick model. **e** Superposition of helix α1 (left), α2 (middle), and α4 (right) from four different structures. The helical movements of α1, α2, and α4 upon substrate binding are drawn in black lines. The key residues, S106 and S191, in the middle and right panels are shown in stick and the side-chain movements are drawn in red lines.

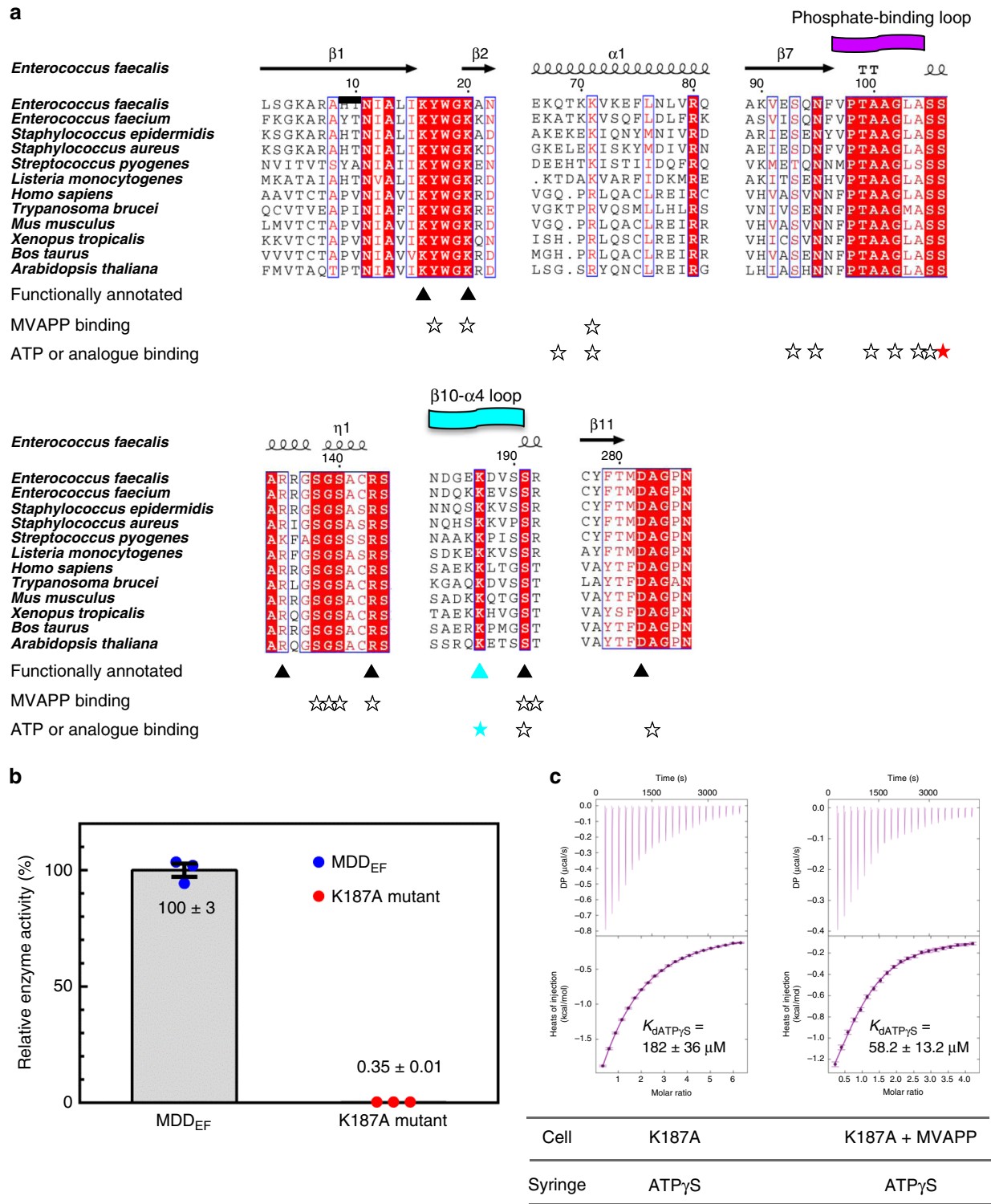

**Fig. 6 The key residue, K187, involved in enzyme catalysis. a** Partial sequence alignment results of the MDD family of proteins (see "Methods" section for details). Identical residues are highlighted in red. Homologous residues have their one-letter codes in red. Residues characterized to be functionally important among MDD proteins are marked by solid triangles[17, 18, 40]. Residues involved in substrate interaction are marked by empty stars and color-filled stars (red: S106; cyan: K187). The phosphate-binding loop and the β10-α4 loop are indicated with curved ribbons in magenta and cyan, respectively. **b** Relative enzyme activity of the K187A mutant and the wild-type MDD$_{EF}$. The enzyme activities of MDD$_{EF}$ and K187A mutant of MDD$_{EF}$ were determined at saturating concentrations of two substrates (MVAPP = 200 μM; MgATP = 800 μM) and normalized by the means of wild-type MDD enzyme activity ($n$ = 3 independent experiments). The means and the standard error (SEM) were derived from each triplicate. Source data are provided as a Source Data file. **c** ITC experiments of MDD$_{EF}$ and the K187A mutant. Left: the K187A mutant (100 μM) was titrated with ATPγS (3 mM). Right: the K187A mutant (100 μM) was pre-incubated with MVAPP (1 mM) and then titrated with ATPγS (2 mM). The thermodynamic parameters of each experiment are listed in Supplementary Table 6. The protein concentration was adjusted to 100 μM and all the protein and titrants were dissolved in the buffer containing 100 mM HEPES, pH 7, 100 mM KCl and 10 mM MgCl$_2$. Error bars in **b** and **c** represent standard error of the mean (SEM).

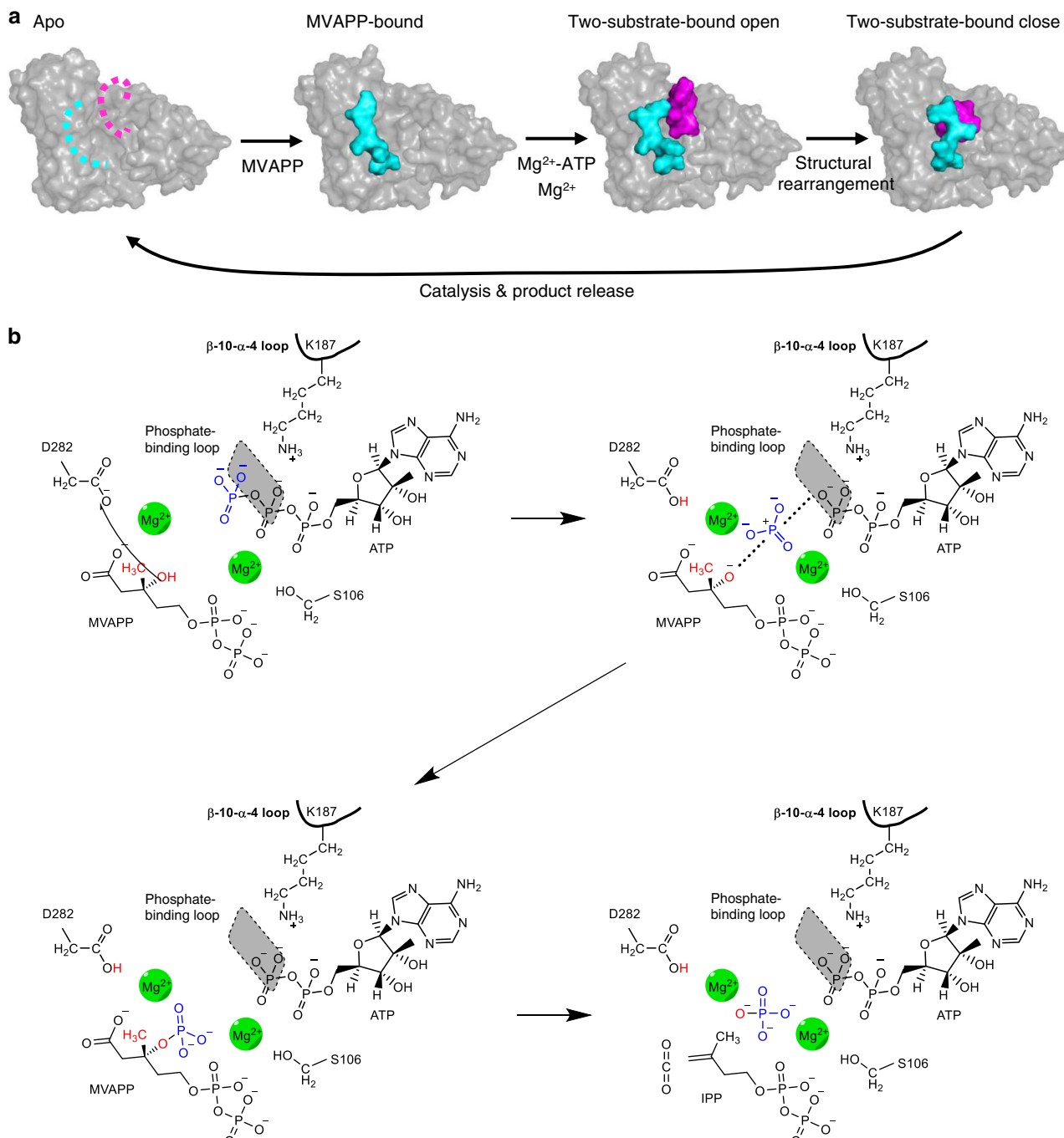

**Fig. 7 Proposed model for the detailed MDD enzyme mechanism. a** Changes in the $MDD_{EF}$ structure upon substrate binding. Left: unbound $MDD_{EF}$. Middle-left: MVAPP-bound $MDD_{EF}$; the β10-α4 loop is presented as cyan surface. Middle-right: an open conformation of $MDD_{EF}$ bound with MVAPP and ATP; the β10-α4 loop and the phosphate-binding loop are shown in cyan and magenta surfaces. Right: a closed conformation of $MDD_{EF}$ bound with both substrates after conformational rearrangement; the β10-α4 loop and the phosphate-binding loop are shown in cyan and magenta surfaces. Enzyme catalysis occurs after these two loops close the active site entrance, followed by product release (isopentenyl diphosphate (IPP), ADP, $CO_2$, and phosphate). **b** Dissociative phosphoryl transfer mechanism of $MDD_{EF}$ during enzyme catalysis. Top-left: D282 is orienting the 3'-OH group of MVAPP (red). Top-right: dissociative phosphoryl transfer occurs and the metaphosphate (blue) is produced. Bottom-left: the metaphosphate attaches to the 3' oxygen (red) of MVAPP and the proton is transferred to the transferred phosphate group. Bottom-right: dephosphorylation and decarboxylation occur and produce products, IPP, ADP, phosphate, and $CO_2$. K187 from the β10-α4 loop and metal ions in the active site are involved in neutralizing the negatively charged environment and assist catalysis. The phosphate-binding loop (shown as a gray shadow as it is in a plane above the rest of the figure) is involved in the ATP binding in the closed form of the $MDD_{EF}$ conformation.

closed conformation, the side-chain of S191 rotates to provide an anchoring point for a transient stay of K187 in the active site. The conserved lysine residue has roles mainly in catalysis. After the chemical steps of catalysis, products (IPP, ADP, $CO_2$, and phosphate) are released from the enzyme and another enzymatic reaction takes place. These findings provide a comprehensive view of structural changes, which link to the specific function of each component orchestrated in the active site of MDD during the reaction.

In other MDD structures, different conformations of the β–α loop (corresponding to the β10-α4 loop in $MDD_{EF}$) and the phosphate-binding loop have been determined. That may be the consequence of the flexibility of these two loops in the MDD family of proteins. In the present structural study, the positions of two loops were determined in the ligand-bound $MDD_{EF}$ structures but not in apo-$MDD_{EF}$. Although it is consistent with our previous study on $MDD_{EF}$ in which a MVAPP-mediated ATP-binding mechanism upon substrate binding was suggested[24], our results are different from other published structures of MDD proteins, in which both or either loops are ordered in the apo structures of MDDs.

The published MDD structures have been compared and summarized in Supplementary Table 1. Among the apo-MDD structures, only the apo-$MDD_{EF}$ (5V2M, 6E2S) and apo-$MDD_{SE}$ (3QT5) structures have no defined electron densities of the β–α loops. It was observed that the presence of the β–α loop in these apo structures is accompanied by the well-defined upcoming α helix and the direct crystal contacts either in the β–α loop or in a region adjacent to the preceding β10 strand (α8 η4 β13 in $MDD_{EF}$, Fig. 1; Supplementary Fig. 7a). In $MDD_{SE}$-MVAPP (4DU7), the posterior α-helix density is well-defined and the density of the β-α loop can also be observed if compared with the apo form of $MDD_{SE}$ (3QT5). This suggests that the binding of MVAPP stabilizes the posterior α-helix and the β-α loop, consistent with our present results. In the structures of MDDs bound with MVAPP, only $MDD_{EF}$ (6E2T) has an undefined phosphate-binding loop. Other MVAPP-bound MDDs have stabilized phosphate-binding loops possibly due to crystal contacts directly involving this loop or the C-terminal insertion covering the corresponding β-strands in eukaryotic MDDs (Supplementary Table 1). In contrast to all other published structures, in the $MDD_{EF}$ structures, there is no crystal contact in the regions described above, which should allow the flexible loops to take up unrestricted conformations. This also may be one of the reasons that different ligand-bound $MDD_{EF}$ structures can be obtained by soaking experiments.

The MDD enzyme mechanism involves phosphoryl transfer of γ-phosphate of ATP to the 3′-oxygen of MVAPP[17,28]. However, how the γ-phosphoryl group of ATP transfers to MVAPP is unclear. In $MDD_{EF}$-MVAPP-ADPBeF$_3$-Mg$^{2+}$, the distance between the phosphoryl donor (Oβ of ADP) and the acceptor (3′-O of MVAPP) is 5.7 Å and BeF$_3^-$ is located in the in-line phosphoryl transfer position (Supplementary Fig. 1). "Pauline bond order" is frequently used for describing a phosphoryl transfer reaction belonging to either an associative or dissociative mechanism in protein kinases or phosphatases[39]. Based on the distance information from the structural model of $MDD_{EF}$-MVAPP-ADPBeF$_3$-Mg$^{2+}$, the bond order was calculated (Supplementary Note 1 and Supplementary Eq. 1) and estimated to be 0.014, corresponding to 1.4% associative and 98.6% dissociative. If the standard coordinate error (0.18 Å)[43,44] in the structure is considered, the bond order in the transition state of phosphoryl transfer could range from 0.007 (99.3% dissociative) to 0.027 (97.3% dissociative). In the active site of the closed complex structure, the ligands were packed tightly, and any movement of MVAPP and/or ADP would result in steric clashes. Thus, our

structural findings suggested that the phosphoryl transfer step in the reaction would likely belong to dissociative phosphoryl transfer, in which metaphosphate ($PO_3^-$) transiently existed during catalysis. These conclusions are also consistent with a recently published QM/MM study in which the active site environment of MDD was generated based on pyruvate kinase (PDB: 3HQP)[45]. The second metal may also be involved in changing the position the γ-phosphate from ATP to approach the 3′-oxygen of MVAPP to facilitate the reaction.

Although the conserved Asp in MDDs (D282 in $MDD_{EF}$) was previously suggested as a general base for the deprotonation of 3′-OH of MVAPP, which then triggers nucleophilic attack on the γ-phosphate of ATP to initiate phosphoryl transfer, a recent study on MDD from Sulfolobus solfataricus ($MDD_{SS}$) has shown that the D281T or D281V mutants produce the 3′-phosphate-MVAPP intermediate but not the final product, IPP. These suggest that the conserved Asp in MDDs may be involved in a later step, as the dephosphorylation and decarboxylation[28]. However, the production of 3′-phosphate-MVAPP by the D281T or D281V mutants is much slower than the production of IPP by the wild-type $MDD_{SS}$, somewhat a puzzling result if the conserved Asp is not at all involved in phosphoryl transfer.

From structural observations, we suggest that this conserved Asp may have multiple roles in MDD catalysis. It has been suggested that the "catalytic" Asp in the active site of kinases could confine the position of the hydrogen of the hydroxyl group of a substrate through hydrogen bond interactions and thus prepare a productive rotamer of the hydroxyl group for phosphoryl transfer[42]. It has also been shown that in a dissociative phosphoryl transfer reaction, the substrate could remain protonated before and during the transition state[46]. In the case of $MDD_{EF}$, the carboxyl group of D282 is 3.3 Å apart from the 3′-oxygen of MVAPP. This suggests that one of the roles of Asp in MDDs is to orient the 3′OH group of MVAPP through hydrogen bonding in a position for an effective phosphoryl transfer. Although a low pH environment would affect the protonation states of catalytic residues and substrates, resulting in a decrease in enzymatic activity[47], MDD enzymes remain active under acidic conditions (above pH 3)[48,49] and D282 of $MDD_{EF}$ under our crystallization conditions (pH 4.6) remains 80% un-protonated (Supplementary Note 1 and Supplementary Eq. 2, assuming the pKa value of the carboxyl group of D282 is 3.9), supporting its role in confining the hydrogen of the 3′-OH group of MVAPP in a certain orientation during dissociative phosphoryl transfer. However, it remains unclear what could serve as a proton acceptor after the formation of the 3′-phosphate-MVAPP intermediate in the case of MDD catalysis, although the transferred phosphate moiety could play that role[28]. In addition, the structural studies on the enzymatically dead D283A mutant of $MDD_{SE}$ (4DPW) showed that the binding pose of MVAPP in the complex structure of $MDD_{SE-D283A}$-MVAPP-ATPγS differs from that in other MDD-ligand-bound structures and the carboxyl group of MVAPP does not interact with its binding partner R144[18]. This suggests that the conserved Asp may have an implied function for preventing non-catalytically binding of MVAPP in the active site. This may also be the case in the D281T or D281V mutants of $MDD_{SS}$.

The current hypothesis for the MDD enzyme mechanism also suggests that the catalytic Asp residue facilitates dephosphorylation/decarboxylation of 3′-phosphate-MVAPP either by changing the conformation of the 3′-phosphate-MVAPP via negative charge repulsion or by stabilizing the carbocation intermediate right after dephosphorylation of the 3′-phosphate-MVAPP[28]. From our crystal structure of $MDD_{EF}$-MVAPP-ADPBeF$_3$-Mg$^{2+}$, large conformational changes of 3′-phosphate-MVAPP might not happen in such packed active site environment. Therefore, stabilizing the carbocation intermediate by the

conserved Asp could be a plausible model, although the influence of metal ions in the transition state remains unclear.

Based on these considerations, we have proposed a possible model for demonstrating the chemical steps of $MDD_{EF}$ catalysis. In the closed complex structure of $MDD_{EF}$ bound with metals, MVAPP and ATP, D282 confine the 3′-OH of MVAPP to prepare a phosphate acceptor (Fig. 7b, top-left); second, metaphosphate transiently exists during the transition state (Fig. 7b, top-right); third, MVAPP receives metaphosphate to produce a 3′-phosphate-MVAPP intermediate (Fig. 7b, bottom-left). Lastly, dephosphorylation and decarboxylation of the intermediate occurs to produce products, IPP, $CO_2$, phosphate, and ADP (Fig. 7b, bottom-right).

In summary, our findings provide detailed information of the $MDD_{EF}$ enzyme mechanism in the aspects of substrate binding and catalysis. These results provide detailed roles for conserved residues and identify the binding sites for two magnesium ions involved in catalysis. Movements in two unrestricted loops and three helices sequentially build the active site and define a potential checkpoint for enzyme activity. Finally, the structures in these different forms will serve as platforms for structure-based drug development, where this work can also be applied to the control of bacterial infections caused by multidrug-resistant *Staphylococci*, *Streptococci*, and *Enterococci*.

## Methods

**Preparation of recombinant $MDD_{EF}$ and the K187A mutant.** A modified site-directed mutagenesis method[50] was used to create the K187A mutant of $MDD_{EF}$[51]. The sequence of the forward primer from 5′ to 3′ is "CTTAATTAATGATGGC GAAGCAGATGTTTCCAGCCGTGATG", and the sequence of the reverse primer is "CATCACGGCTGGAAACATCTGCTTCGCCA TCATTAATTAAG". The 50 μl PCR solution contained the forward and reverse primers (1 μM, respectively), dNTP (200 μM of each), Phusion HF buffer (1×), template DNA (0.1–1 μl), DMSO (2%), Phusion DNA polymerase (one unit) and sterile water. The PCR program was set to be 1 cycle of denaturation (95 °C, 1 min), 25 cycles of the three-step reaction (1. Denaturation, 95 °C, 30 s; 2. Annealing, 62 °C, 30 s; 3. Extension, 72 °C, 5.5 min) and one cycle of the final extension (72 °C, 10 min). After PCR, the original templates containing methylated DNA were digested by Dpn1 (1 μl) for 1 h at 37 °C. The K187A mutant construct was transformed into *Escherichia coli* BL21 (DE3, Novagen). Transformed cells were cultured in LB broth (containing 50 mg ml⁻¹ kanamycin) at 37 °C to an $A_{600nm}$ value of 0.8–1.0. The protein expression and purification procedures for the K187A mutant were similar to the procedures for obtaining wild-type $MDD_{EF}$ proteins[24]. Protein expression of K187A was conducted by adding isopropyl 1-thio-β-D-galactopyranoside (IPTG) (0.1 mM) to the bacterial culture for 4 h at 37 °C. Cells were harvested by centrifugation at 9605 g, resuspended in binding buffer (50 mM sodium phosphate at pH 7.4, 300 mM NaCl, and 10 mM imidazole), and lysed to homogeneity by French Press. His-tagged K187A proteins were trapped in a Ni²⁺-NTA column followed by elution with an increasing percentage of elution buffer (50 mM sodium phosphate at pH 7.4, 300 mM NaCl, and 300 mM imidazole). Eluted fractions were examined by SDS-PAGE and fractions containing K187A were collected and dialyzed against dialysis buffer (25 mM Tris-HCl, pH 8.0, 100 mM NaCl, and 10 mM MgSO₄) for two times, one with β-mercaptoethanol (β-ME) (20 mM) and the other one without β-ME. The N-terminal His-tag was removed from K187A by recombinant tobacco etch virus (TEV) protease treatment in dialysis buffer containing 1 mM DTT and 0.5 mM EDTA, followed by dialysis against the dialysis buffer without DTT and EDTA. His-tagged TEV and residual His-tagged K187A were removed by passing the protein mixture through a nickel affinity resin. Purified K187A protein solution was concentrated to 8–10 mg ml⁻¹ by ultrafiltration and stored at −20 °C or −80 °C for long-term storage.

**Enzymatic activity of wild-type $MDD_{EF}$ and the K187A mutant.** The $K_{mMVAPP}$ value and the $K_{mATP}$ value of $MDD_{EF}$ were ~40 and ~160 μM[24], respectively. Enzymatic reactions for wild-type $MDD_{EF}$ and the K187A mutant were performed at saturated substrate concentrations (MVAPP = 200 μM, ATP = 800 μM). The enzymatic activities of $MDD_{EF}$ and the K187A mutant were determined using an ATP/NADH enzyme-coupled assay. Each reaction was performed at 30 °C under the conditions (100 mM HEPES, pH 7.0, 100 mM KCl, 10 mM MgCl₂, 0.2 mM NADH, 0.4 mM phosphoenolpyruvate, 4 units of pyruvate kinase, 4 units of lactate dehydrogenase, and 100 nM $MDD_{EF}$[18] or 1 μM K187A). Initial velocity of each reaction was determined and relative enzymatic activity of the K187A mutant was calculated by dividing the enzymatic velocity of the K187A mutant by the enzymatic velocity of wild-type $MDD_{EF}$. Each assay has a final volume of 200 μl. Data analysis was performed using SigmaPlot verion 12.5 and GraphPad Prism 6.0.

**Sequence alignment and structural annotation.** The sequences of MDD proteins from organisms (*Enterococcus faecalis*; *Enterococcus faecium*; *Staphylococcus epidermidis*; *Staphylococcus aureus*; *Streptococcus pyogenes*; *Listeria monocytogenes*; *Homo sapiens*; *Trypanosoma brucei*; *Mus musculus*; *Xenopus tropicalis*; *Bos taurus*; *Arabidopsis thaliana*) were aligned using EBI Clustal Omega[52]. The secondary structure elements were drawn using ESPript3.0[53] based on the structural model of $MDD_{EF}$ in complex with MVAPP, ADP, cobalt, and sulfate ($MDD_{EF}$-MVAPP-ADP-$SO_4^{2-}$-$Co^{2+}$) in this study.

**Preparation of crystals of apo and ligand-bound $MDD_{EF}$.** $MDD_{EF}$ was crystallized using the sitting drop method with 1.6 M ammonium sulfate, 50 mM sodium acetate, pH 4.6, as established in our previously published results[24]. Buffer exchange procedures for ligand soaking experiments were performed by replacing the crystallization buffer with soaking buffer (26% PEG3350, 5 mM MgCl₂, 50 mM sodium acetate, pH 4.6) and crystals were then equilibrated in soaking buffer for 10 min. Each ligand was dissolved in soaking buffer to a final concentration of 2 mM and a small amount of ligand solution (0.12–0.15 μl) was added into the drop for a one-day soaking procedure. For cryo-protection, dehydration buffer (30% PEG3350, 15% PEG400, 50 mM sodium acetate, pH 4.6) was placed into the bottom well of each individual chamber in order to increase the PEG3350 concentration in the sitting drop and left for one day. Crystals of $MDD_{EF}$ soaked with ligands and metal ions were labeled in a manner of "$MDD_{EF}$-ligand-metal". Crystals with or without ligands ($MDD_{EF}$-$SO_4^{2-}$, $MDD_{EF}$-MVAPP, $MDD_{EF}$-MVAPP-ADP-$SO_4^{2-}$-$Co^{2+}$, $MDD_{EF}$-MVAPP-AMPPCP-$Mg^{2+}$, and $MDD_{EF}$-MVAPP-ADPBeF₃-$Mg^{2+}$) were obtained and frozen in liquid nitrogen.

**Data collection and structure analysis.** The diffraction data of an apo form of $MDD_{EF}$ ($MDD_{EF}$-$SO_4^{2-}$) and $MDD_{EF}$ soaked with ligands ($MDD_{EF}$-MVAPP, $MDD_{EF}$-MVAPP-ADP-$SO_4^{2-}$-$Co^{2+}$ and $MDD_{EF}$-MVAPP-ADPBeF₃-$Mg^{2+}$) were collected at the 23-ID-B and 23-ID-D beamline at Advanced Photon Source (APS) at Argonne National Laboratory in Chicago. Diffraction data of two ligand-bound $MDD_{EF}$, $MDD_{EF}$-MVAPP-AMPPCP-$Mg^{2-}$ and $MDD_{EF}$-MVAPP-ADP-$SO_4^{2-}$-$Co^{2+}$ (for cobalt anomalous signal data) were collected on the home-source X-ray diffraction equipment (Purdue University Macromolecular Crystallography Facility). The HKL2000 software was used for space group determination, data integration, data reduction and data scaling[54], and after data processing, a scale-packed reflection file (.sca) was generated. The software in CCP4, scalepack2mtz, was then used to convert the scalepack reflection file (.sca) to an MTZ format (.mtz) with R-free flag assigned (5–5.5%)[55].

The program phenix.phaser was used to estimate the phases based on the molecular replacement method[56]. Using the previous deposited $MDD_{EF}$-ATP structure (5V2L) as a search model. One solution was found with rotation function log-likelihood gain (LLG) > 0 and translation function Z-score (TFZ) > 8. Structure refinement was performed using phenix.refine[56]. Model geometry (*XYZ* coordinates), atomic positions (Real-space), and atomic B-factors (individual B-factors) were refined, and models were manually examined in the graphical program Coot[57] based on the electron density map ($2F_o − F_c$) and the difference map ($F_o − F_c$). The simulated-annealing (Cartesian) option was employed in the first few refinement runs. The crystallographic information file (.cif) and the PDB format file (.pdb) of ligands (ADP, AMPPCP, MVAPP, $Co^{2+}$, BeF₃⁻) were generated using the program phenix.eLBOW[56]. After a few runs of structure refinement without ligands, ligands were manually placed and fitted into the weighted difference electron density maps ($F_o − F_c$) in Coot[57]. Ligand-omit maps were generated using the phenix.composit_omit_map software for evaluating structural models of ligands. Each ligand-omit map and its corresponding structure were depicted in pymol[58]. Target function optimization (Optimize X-ray/stereochemistry weight, Optimize X-ray/ADP weight) was also chosen for optimizing the weight between the X-ray data and the structural model. Water molecules were built and inspected in Coot.

**Isothermal titration calorimetry experiments.** The preparation of TEV-treated K187A was described above. The protein solution was dialyzed against buffer which is the same as used in the enzymatic reactions described previously (100 mM HEPES, pH 7, 100 mM KCl, and 10 mM MgCl₂). All the buffer solutions in ITC experiments were filtered through a 0.45 μM filter and degassed for 1 h at room temperature. The protein concentration was adjusted to 100 μM (260 μl). Each ligand (MVAPP, ATP, and ATPγS) was prepared in the same dialysis buffer to avoid buffer mismatch. The concentration of each titrant was optimized in different experiments based on the experimental designs simulated using the MicroCal Origin 7.0 software package, and the final concentration of each ligand was adjusted appropriately. The ITC instrument, MicroCal iTC200, was employed for isothermal titrations in this study and the reference cell was filled with ddH₂O containing 0.01% sodium azide. The experimental temperature was set at 25 °C. Each experimental profile was composed of the addition of an initial aliquot of 0.4 μl, followed by 22 aliquots of 1.8 μl of the substrate or ligand solution. The time interval between two consecutive injections was 180 s. The data were further processed with NITPIC[59] and analyzed using a one-site model in SEDPHAT[60]. Figures were generated using GUSSI in SEDPHAT.

**Reporting summary**. Further information on research design is available in the Nature Research Reporting Summary linked to this article.

## Data availability

The X-ray data and corresponding atomic coordinates of $MDD_{EF}$ at unbound and different bound states ($MDD_{EF}$-$SO_4^{2-}$: 6E2S; $MDD_{EF}$-MVAPP: 6E2T; $MDD_{EF}$-MVAPP-AMPPCP-$Mg^{2+}$: 6E2U; $MDD_{EF}$-MVAPP-ADPBeF$_3$-$Mg^{2+}$: 6E2V) and data for cobalt anomalous dispersion experiments (synchrotron data for $MDD_{EF}$-MVAPP-ADP-$SO_4^{2-}$-$Co^{2+}$: 6E2W; home-source data for $MDD_{EF}$-MVAPP-ADP-$SO_4^{2-}$-$Co^{2+}$: 6E2Y) have been deposited in the Protein Data Bank. Source data are provided with this paper.

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

## Acknowledgements

We thank Tim Schmidt (Macromolecule Crystallography X-Ray Diffraction Lab, Purdue University) for his help with X-ray data collection and the Advanced Photon Source (APS) at Argonne National Laboratory in Chicago for access to the beamlines of 23-ID-B and 23-ID-D. GM/CA@APS has been funded in whole or in part with Federal funds from the National Cancer Institute (ACB-12002) and the National Institute of General Medical Sciences (AGM-12006). This research used resources of the Advanced Photon Source, a U.S. Department of Energy (DOE) Office of Science User Facility operated for the DOE Office of Science by Argonne National Laboratory under Contract No. DE-AC02-06CH11357. The Eiger 16M detector was funded by an NIH–Office of Research Infrastructure Programs, High-End Instrumentation Grant (1S10OD012289-01A1). We also gratefully acknowledge use of the Macromolecular Crystallography Shared Resource with support from the Purdue Center for Cancer Research and NIH grant P30 CA023168.

## Author contributions

C.-L.C. and C.V.S. designed the experiments. J.C.M cloned the MDD protein. C.-L.C. purified the K187A mutant and the wild-type MDD$_{EF}$ proteins for study on enzymology, thermodynamics, and crystallization. C.-L.C. analyzed data. L.N.P. designed and performed the ITC experiments. C.N.S. assisted in reviewing the MDD X-ray crystal structures. C.-L.C. and C.V.S. collaborated in writing the article.

## Competing interests

The authors declare no competing interests.
