## [Peer Review File · Nature Communications]

Reviewers' comments:

Reviewer #1 (Remarks to the Author):

The manuscript "Visualizing the enzyme mechanism of mevalonate diphosphate decarboxylase" by C.-L. Chen et al. reports the structural studies on mevalonate decarboxylase from a vancomycin resistant enterococcus, *Enterococcus faecalis* (MDDEF). The authors succeeded in solving not only the apo-structure but also the structures with the substrate mevalonate diphosphate (MVAPP), MVAPP and an ATP analog AMPPCP, MVAPP and ADP, or MVAPP and ADP/BeF₃⁻. As observed with the structures of other MDDs, the MDDEF structures took open or closed conformations, which allowed the authors to suppose the reaction mechanisms of the enzyme through substrate binding and phosphoryl-transfer, which is the first half of the MDD reaction. It is quite impressive that the authors could verify the binding positions of magnesium ions in the catalytic center of the enzyme. To my knowledge, this is the first discovery for this enzyme family and also the information needed to reveal detailed catalytic mechanism of MDDEF and to develop inhibitor molecules specific to the enzyme. Because human also possesses MDD, only such specific inhibitors can be utilized as antibiotics for VREs. The metal binding sites are confirmed with anomalous dispersion observed in the analysis of the crystals soaked with MVAPP and ADP in the presence of cobalt ion, instead of magnesium ion. This study is basically an extension of their previous studies on the structures and biochemical properties of MDDEF, but it clearly made a substantial progress. The manuscript is readable, and all the data look sound and persuasive. The reviewer, however, still has several concerns with the manuscript. First, the authors proposed the substrate binding mechanism based on the open-close change observed in the enzyme structures through the bindings of different substrates or their analogs. The proposed mechanism fits well the structural changes observed with MDDEF, but not with other MDD structures previously reported. For example, the alpha10-beta4-loop was in the closed position in the MDDEF structure with MVAPP/ADP/BeF₃⁻/Mg²⁺, but in the open position with MVAPP/AMPPCP/Mg²⁺, thus the authors concluded that the former structure represents the pre-phosphoryl transfer state, while the latter does an earlier state. In contrast, the loop, and also the Lys residue corresponding to Lys187, were reported to be at the same positions in MDD from an archaeon *Sulfolobus solfataricus* (MDDSS) not only in similar substrate-binding structures but also in its apo form (pdb: 5gmd, 5gme, 4z7c). The discrepancy casts question if the induced-fit mechanism proposed in this paper is general for MDDs or not. Additional discussion based on the comparison with the previously reported MDD structures is needed. Secondly, the mechanism of the ATP-dependent decarboxylation of MVAPP described in the manuscript, in which Asp282 acts as a base that deprotonate 3'-OH of MVAPP, has been controverted by recent finding of a MDD-homolog kinase that lacks the corresponding Asp residue but can catalyze 3'-phosphorylation of mevalonate (Vinokur J.M. et al., 2015, Protein Sci.), and also by a mutagenic study on MDDSS, which led the proposal of a hypothesis that Asp residue is required in the decarboxylation step (Motoyama K. et al., 2017, J. Biol. Chem.). If the authors want to refute the new hypothesis, they should perform QM/MM calculation study based on the structure of MDDEF binding MVAPP/ADP/BeF₃⁻/Mg²⁺ or MVAPP/AMPPCP/Mg²⁺, which will clarify the protonation states of the substrates and catalytic residues and might support the less-persuasive discussion in page 15-16.

Minor concerns:

- 1) Page 2, line 7: large conformational "change"?
- 2) Page 3, the 1st paragraph: The MEP pathway is found primarily in bacteria, not in prokaryotes. Archaea, another prokaryotes lineage, have the MVA pathway.
- 3) Page 3, line 8: privarily > primarily
- 4) Page 17, lines 19-: The method utilized for site-directed mutagenesis is obviously based on the QuickChange method. Appropriate references such as Zheng L. et al. (2004) Nucleic Acid Res. are needed.
- 5) Page 19, line 10: Instead of 100 nM MDDEF, a higher amount of the mutant enzyme was considered to be used. Please show the amount.
- 6) Page 21, lines 19-20: ligands "were" manually placed

7) Supplementary Table 4: The table seems not showing the angles of alfa-helices, but their 3D positions. I am afraid that the title and legend are confused with those in Supplementary Table 3.

Reviewer #2 (Remarks to the Author):

In the manuscript entitled "Visualizing the Enzyme Mechanism of Mevalonate Diphosphate Decarboxylase" by Chen et al. (Manuscript NCOMMS-19-27996-T) the authors describe that two Mg²⁺ ions are bound to the enzyme preloaded with substrate and ATP. By this sequential binding, the active site is shaped for catalysis.

The authors used X-ray crystallography to visualize the Mg²⁺ ions plus structures with transition state analogues. The resolution of the diffraction data is very high and the quality of the refined structures is very good. Surprised by the localization of the two Mg²⁺ ions, the authors generated structure-based variants of the MDD protein.

To the reviewer's point of view the current version of the manuscript is very well written and comprehensive. Nevertheless, I have the impression that most of the techniques described were already very well established in the lab and for instance described in earlier publications (Chen et al. (2017) JBC. For instance, the crystallization system has been described in the JBC publication. Probably in many instances, the described methods could be drastically shortened. Most of the designed variants have already been shown, to have functional implications. It is not proven, that the proposed mechanism is valid in other organisms than *S. faecalis* as well. In particular, the beta10-alpha4 loops show very little conservation. What is the function of R192? It seems to be conserved in bacteria, where as in higher organisms it seems to be a strict threonine residue. Is the catalytic side in both monomers identical?

Minor points:

- Throughout the manuscript, there is a mixture of one-letter and three-letter code for amino acids.
- Page 5, line 13: Space is missing 1.6 M
- Page 7, line 7: "extra density". Please be more precise "extra electron density"
- Page 7, line 19: probably "...next section" is meant
- Page 19, line 8: Space is missing after 30
- Page 19, line 18: please remove dot after Mus
- Figure 1, panel a: I do not think that the topology diagram is necessary. The overall structure of MDDs is very well known
- Figure 6, panel a: please carefully check the numbering of the alignment. I see some inconsistencies between the numbers of amino acids mentioned in the body of the text the provided numbering in the figure. Names of organism should be in italic.

Reviewers' Comments with Responses:

Reviewer #1 (Remarks to the Author):

The manuscript “Visualizing the enzyme mechanism of mevalonate diphosphate decarboxylase” by C.-L. Chen et al. reports the structural studies on mevalonate decarboxylase from a vancomycin resistant enterococcus, *Enterococcus faecalis* (MDDEF). The authors succeeded in solving not only the apo-structure but also the structures with the substrate mevalonate diphosphate (MVAPP), MVAPP and an ATP analog AMPPCP, MVAPP and ADP, or MVAPP and ADP/BeF₃⁻. As observed with the structures of other MDDs, the MDDEF structures took open or closed conformations, which allowed the authors to suppose the reaction mechanisms of the enzyme through substrate binding and phosphoryl-transfer, which is the first half of the MDD reaction. It is quite impressive that the authors could verify the binding positions of magnesium ions in the catalytic center of the enzyme. To my knowledge, this is the first discovery for this enzyme family and also the information needed to reveal detailed catalytic mechanism of MDDEF and to develop inhibitor molecules specific to the enzyme. Because human also possesses MDD, only such specific inhibitors can be utilized as antibiotics for VREs. The metal binding sites are confirmed with anomalous dispersion observed in the analysis of the crystals soaked with MVAPP and ADP in the presence of cobalt ion, instead of magnesium ion. This study is basically an extension of their previous studies on the structures and biochemical properties of MDDEF, but it clearly made a substantial progress. The manuscript is readable, and all the data look sound and persuasive. The reviewer, however, still has several concerns with the manuscript.

Question 1:

First, the authors proposed the substrate binding mechanism based on the open-close change observed in the enzyme structures through the bindings of different substrates or their analogs. The proposed mechanism fits well the structural changes observed with MDDEF, but not with other MDD structures previously reported. For example, the alpha10-beta4-loop was in the closed position in the MDDEF structure with MVAPP/ADP/BeF₃⁻/Mg²⁺, but in the open position with MVAPP/AMPPCP/Mg²⁺, thus the authors concluded that the former structure represents the pre-phosphoryl transfer state, while the latter does an earlier state. In contrast, the loop, and also the Lys residue corresponding to Lys187, were reported to be at the same positions in MDD from an archaeon *Sulfolobus solfataricus* (MDDSS) not only in similar substrate-binding structures but also in its apo form (pdb: 5gmd, 5gme, 4z7c). The discrepancy casts question if the induced-fit mechanism proposed in this paper is general for MDDs or not. Additional discussion based on the comparison with the previously reported MDD structures is needed.

Authors' Response to Question 1:

We believe the results outlined in this manuscript are relevant to the mechanism of all MDDs for a number of reasons. One comes from earlier enzymology experiments as published in Chen et al. (2017). The substrate binding mechanism of MDD_{EF} was determined enzymatically in this previous study¹, in which MVAPP is the first substrate and the enzymatic mechanism was determined as a sequential ordered bi-substrate mechanism. These experimental results were completely consistent with the parallel enzymology study of MDD from chicken². The comparable results from these MDD proteins across domains suggest that the enzyme mechanism of MDD is conserved.

Starting from that premise, we then sequentially soaked MDD crystals with MVAPP and ATP analogs (AMPPCP or ADPBeF₃) in order to compare the changes in the enzyme-substrate complexes trapped at different states in the reaction mechanism. The results led us to propose a model describing induced-fit conformational changes of flexible loops in MDD upon substrate/cofactor binding. The induced fit binding was also supported by ITC studies on the reaction in our previous publication.

The reason we are able to trap these states involves the particular crystal form in which MDD_{EF} crystallized. In order to have the cleanest view of changes at the active site, it is optimal to have this portion of the enzyme facing solvent gaps in the crystal and to not be involved in any crystal contacts. Our crystals of MDD_{EF} have this unique advantage that allows the enzyme to freely catalyze the reaction. We therefore were able to soak in substrates and cofactor combinations that lead to intermediate states in the reaction.

We have now completed full structure comparisons for MDD structures in the PDB from which we conclude that the loops we describe as being involved in the mechanism are involved in crystal contacts and are restrained in their positions. It is interesting that the loop conformations in other MDD structures mimic the changes we observe on substrate binding in MDD_{EF}. In these MDD structures, conformations of the β 10- α 4 loop and the phosphate binding loop (herein described as the P-loop) have been determined in a variety of states, but with the same general features. As is found in our structures, the β 10- α 4 loops in other MDD structures appear near the active site along with the movement of α 8- η 4- β 13 bending toward and the α 4 helix moving closer to the active site (compared with apo-form MDD_{EF}.) Similarly, the P-loop also appears to be well defined in some of these structures along with a rearrangement of α 2 and β 7. These structures suggested that crystal packing directly on the two loops or their flanking regions (α 8- η 4- β 13 and α 4 for the β 10- α 4 loop; α 2 and β 7 for the P-loop) may well stabilize the loops in one of their potential configurations during the catalytic cycle, which is then selected from a set of conformations freely sampled in our crystals where this restriction is not in place. Therefore we feel it is probable that the loop configurations in these structures are stabilized during crystallization.

To describe this, we have provided new supplementary information in Supplementary

Table 1 and Supplementary Figure 7 to compare the crystallization conditions and the loop conformations in the published PDB structures. We have also covered these new comparisons in the Discussion section of the manuscript to describe how we understand the flexibility of the two loops of MDD in the apo or ligand-bound conformations at this point in MDD centered research.

Changes that address this question can be found, highlighted in yellow, in the manuscript as outlined by page number below:

Main Page 16-17

Supplementary Page 8

Supplementary Page 9

Question 2

Secondly, the mechanism of the ATP-dependent decarboxylation of MVAPP described in the manuscript, in which Asp282 acts as a base that deprotonate 3'-OH of MVAPP, has been controverted by recent finding of a MDD-homolog kinase that lacks the corresponding Asp residue but can catalyze 3'-phosphorylation of mevalonate (Vinokur J.M. et al., 2015, Protein Sci.), and also by a mutagenic study on MDDSS, which led the proposal of a hypothesis that Asp residue is required in the decarboxylation step (Motoyama K. et al., 2017, J. Biol. Chem.). If the authors want to refute the new hypothesis, they should perform QM/MM calculation study based on the structure of MDDEF binding MVAPP/ADP/BeF₃-/Mg²⁺ or MVAPP/AMPPCP/Mg²⁺, which will clarify the protonation states of the substrates and catalytic residues and might support the less-persuasive discussion in page 15-16.

Authors' Response to Question 2:

Reviewer #1 has brought to our attention important experimental information that allows us to refine our interpretation of the MDD_{EF} mechanism. We have come to the conclusion that the conserved Asp (D282 in MDD_{EF}) may have multiple functions. First, from the references recommended it can be seen that the MDD_{SE} D283A mutant (4DPW) is considered a dead mutation. Compared with other MDD-MVAPP bound structures, the complex structure of D283 bound with MVAPP and ATP_γS showed that the carboxyl group of MVAPP does not interact with R144 and has an alternative binding configuration in the active site. This suggested that D282 may have an implied function for preventing non-catalytically binding of MVAPP in the active site³.

In light of these experiments, a current analysis of our results suggest a dissociative phosphoryl transfer mechanism during the reaction, in which the conserved Asp residue may function to confine the orientation of the hydroxyl group but not serve as the residue responsible for deprotonation of the 3'-OH group of MVAPP, as suggested earlier in the

literature⁴. However, it is still somewhat unclear from all the MDD structures what is the proton acceptor when 3'-phosphate-MVAPP is formed. The best candidate is the transferred phosphate group, as in the MDDss paper where the authors proposed a protonated 3'-phosphate-MVAPP in the intermediate state⁵. Supporting these conclusions is also a recently published paper with QM/MM studies on a model of MDD, based on the pyruvate kinase structure, which also ruled out the conserved Asp as a proton acceptor⁶. There is another likely possibility for the role of the conserved Asp residue in the stabilization of the carbocation intermediate of the reaction. Since our closed form of MDD_{EF} represents the structure in a pre-phosphoryl transfer state, we do not directly observe this step in the mechanism.

Based on the reasons described above, we have modified the reaction mechanism in Figure 7b to be consistent with the analysis above and have removed the step with deprotonation by the Asp sidechain. Given the very insightful comments by this reviewer on the mechanism, we believe we now present a more consistent view of the MDD reaction. Since we have discarded the contested hypothesis and are no longer in disagreement with other experiments, we now think that QM/MM studies are unnecessary.

Changes that address this question can be found, highlighted in yellow, in the manuscript as outlined by page number below:

Main Page 4, Line 10-22

Main Page 17-20

Main, Page 42, Fig. 6 panel B

Minor concerns:

1) Page 2, line 7: large conformational “change”?

Authors' response: Main Page 2, line 7

2) Page 3, the 1st paragraph: The MEP pathway is found primarily in bacteria, not in prokaryotes. Archaea, another prokaryotes lineage, have the MVA pathway.

Authors' response: Main Page 3, line 9

3) Page 3, line 9: privarily > primarily

Authors' response: Main Page 3, line 9

4) Page 17, lines 19-: The method utilized for site-directed mutagenesis is obviously based on the QuickChange method. Appropriate references such as Zheng L. et al. (2004) Nucleic Acid Res. are needed.

Authors' response: Main Page 20, line 14

5) Page 19, line 10: Instead of 100 nM MDD_{EF}, a higher amount of the mutant enzyme was considered to be used. Please show the amount.

Authors' response: Main Page 21, Line 19

6) Page 21, lines 19-20: ligands “were” manually placed

Authors' response: Main Page 24, line 5

7) Supplementary Table 4: The table seems not showing the angles of alpha-helices, but their 3D positions. I am afraid that the title and legend are confused with those in Supplementary Table 3.

Authors' response: Supplementary Page 14 and Supplementary Page 15

Reviewer #2 (Remarks to the Author):

In the manuscript entitled "Visualizing the Enzyme Mechanism of Mevalonate Diphosphate Decarboxylase" by Chen et al. (Manuscript NCOMMS-19-27996-T) the authors describe that two Mg²⁺ ions are bound to the enzyme preloaded with substrate and ATP. By this sequential binding, the active site is shaped for catalysis.

The authors used X-ray crystallography to visualize the Mg²⁺ ions plus structures with transition state analogues. The resolution of the diffraction data is very high and the quality of the refined structures is very good. Surprised by the localization of the two Mg²⁺ ions, the authors generated structure-based variants of the MDD protein.

To the reviewer's point of view the current version of the manuscript is very well written and comprehensive. Nevertheless, I have the impression that most of the techniques described were already very well established in the lab and for instance described in earlier publications (Chen et al. (2017) JBC).

Question 1

For instance, the crystallization system has been described in the JBC publication. Probably in many instances, the described methods could be drastically shortened.

Authors' response: Main Page 21, Line 4 and Main Page 22, Line 11-14

Question 2

Most of the designed variants have already been shown, to have functional implications. It is not proven, that the proposed mechanism is valid in other organisms than *E. faecalis* as well. In particular, the beta10-alpha4 loops show very little conservation. What is the function of

R192? It seems to be conserved in bacteria, where as in higher organisms it seems to be a strict threonine residue. Is the catalytic side in both monomers identical?

Authors' Response to Question 2:

Using comparisons of the crystal structures of MDDs (Supplementary Table 1), it cannot be shown if other MDD proteins follow the identical conformational changes upon substrate binding due to limitations on many aspects such as crystal conditions, crystal packing, usage of substrate/ligands, etc. Except for our structures soaked with ligands, all of others were co-crystal structures, which may be difficult to compare with each other given the variation in crystallization conditions. The crystal packing of MDD_{EF} is somewhat unique because the flexible loops are not involved in the contact regions and appear to be intrinsically disordered / flexible in the apo structure. This allows us to perform soaking experiments with substrates and cofactors and see structural changes within the same kind of crystals under almost identical conditions. Because other species of MDD structures do not contain all the states which are shown in this study, we have revised the discussion to focus on MDD_{EF} in terms of the mechanism.

The active site cleft and the phosphate binding loop of MDDs are highly conserved⁷. Although the b10-a4 loops share little similarity, the key lysine residue (K187 in MDD_{EF}) is conserved in MDDs across species. From our study, the lysine is mainly involved in catalysis and moderately in substrate binding (interacting with the β - γ -bridging oxygen). In the bacterial structures the function of R192 appears to be in MVAPP binding. This residue is conserved in prokaryotes but replaced with threonine in higher organisms. However in higher organisms, R78 in the α 1 helix and T217 in the a4 of MDD from *Arabidopsis thaliana* (PDB:6N10) interact with the pyrophosphate tail of MVAPP. Similarly, K71 in the α 1 helix and R192 in the α 4 helix of MDD_{EF} interact with MVAPP. The “positional coincidence” of Arg sidechain suggesting evolutionary convergence has been discussed in the recent published paper⁸. We have now discussed this point and provide the citation in the manuscript as well.

Changes that address this question can be found, highlighted in yellow, in the manuscript as outlined by page number below:

Main, Page 12, line 11-14

Main, Page 15 line 23, Page 16 line 1-23, Page 17 line 1

Main, Page 34 Fig. 7

Supplementary Page 8 Supplementary Figure 7

Supplementary Page 9 Supplementary Table 1

Minor points:

1. Throughout the manuscript, there is a mixture of one-letter and three-letter code for amino acids

Authors' response: Changes in entire manuscript

2. Page 5, line 13: Space is missing 1.6 M

Authors' response: Main Page 5, line 19

3. Page 7, line 7: “extra density”. Please be more precise “extra electron density”

Authors' response: Main Page 7, line 12

4. Page 7, line 19: probably “...next section” is meant

Authors' response: Main Page 7, line 23

5. Page 19, line 8: Space is missing after 30

Authors' response: Main Page 21, line 17

6. Page 19, line 18: please remove dot after Mus

Authors' response: Main Page 22, line 3

7. Figure 1, panel a: I do not think that the topology diagram is necessary. The overall structure of MDDs is very well known

Authors' response: Main Page 36 Fig. 1

8. Figure 6, panel a: please carefully check the numbering of the alignment. I see some inconsistencies between the numbers of amino acids mentioned in the body of the text the provided numbering in the figure. Names of organism should be in italic.

Authors' response: Changes in the entire manuscript and specifically in Main Page 41 Fig. 6

1. Chen, C.L., Mermoud, J.C., Paul, L.N., Steussy, C.N. & Stauffacher, C.V. Mevalonate 5-diphosphate mediates ATP binding to the mevalonate diphosphate decarboxylase from the bacterial pathogen. *J Biol Chem* **292**, 21340-21351 (2017).
2. Jabalquinto, A.M. & Cardemil, E. Substrate binding order in mevalonate 5-diphosphate decarboxylase from chicken liver. *Biochim Biophys Acta* **996**, 257-9 (1989).
3. Barta, M.L., McWhorter, W.J., Mizioro, H.M. & Geisbrecht, B.V. Structural Basis for Nucleotide Binding and Reaction Catalysis in Mevalonate Diphosphate Decarboxylase. *Biochemistry* **51**, 5611-5621 (2012).
4. Adams, J.A. Kinetic and catalytic mechanisms of protein kinases. *Chem Rev* **101**, 2271-90 (2001).
5. Motoyama, K. et al. A Single Amino Acid Mutation Converts (R)-5-Diphosphomevalonate Decarboxylase into a Kinase. *J Biol Chem* **292**, 2457-2469 (2017).
6. McClory, J., Hui, C., Zhang, J. & Huang, M. The phosphorylation mechanism of mevalonate diphosphate decarboxylase: a QM/MM study. *Org Biomol Chem* (2019).
7. Barta, M.L. et al. Crystal Structures of *Staphylococcus epidermidis* Mevalonate Diphosphate Decarboxylase Bound to Inhibitory Analogs Reveal New Insight into Substrate Binding and Catalysis. *Journal of Biological Chemistry* **286**, 23900-23910 (2011).
8. Thomas, S.T., Louie, G.V., Lubin, J.W., Lundblad, V. & Noel, J.P. Substrate Specificity and Engineering of Mevalonate 5-Phosphate Decarboxylase. *ACS Chem Biol* **14**, 1767-1779 (2019).

REVIEWERS' COMMENTS:

Reviewer #1 (Remarks to the Author):

I am totally satisfied with the revisions made by the authors, in which all my comments have been addressed. Now the reaction mechanism of MDD proposed by the authors is updated appropriately and agrees well with recent knowledges of the enzyme. The comparison of the structures of MDD solved in the study with the previously known structures of the enzyme and its orthologs, considering the effect of crystal packing, makes sense. I still hope the catalytic mechanism of this interesting decarboxylase is elucidated by QM/MM study, but it is not necessary in this paper. The elucidation will be made in future probably by the authors or other groups.

Reviewer #2 (Remarks to the Author):

In the manuscript entitled "Visualizing the Enzyme Mechanism of Mevalonate Diphosphate Decarboxylase" by Chen et al. (Manuscript NCOMMS-19-27996A) the authors submitted a revised version of their manuscript.

The revised version has substantially improved.

(1) The structural comparison of different MDD structures available in the PDB improves the understanding of the differences of the structures for example different crystal packing effects and hence restricted conformational flexibility of the elements. See also the novel Supplementary Figure 7. Here, I just have a minor point: I personally find it useful if the PDB-ID for the displayed structure (panel c to e) is provided and/or a reference to the primary citation. This suggestion applies as well to Supplementary figures.

The functional implications are now much better to follow. In particular, the discussion of the putative function of Asp282 is now better to understand. By the discussion of the other two recent publications, as suggested by the other reviewer, the authors were able to discard the contested hypothesis section on the mechanism. It is a very beautiful example how structural biology can provide insightful structural information to elucidate an enzymatic mechanism.

I recommend to accept the current version of the manuscript.

Minor points:

- Page 12, line 11: Names of organism should be in italic
- Supplementary Figure 5: I suggest deleting the primer sequences. They are given in the Materials and Methods section.

REVIEWERS' COMMENTS:

Reviewer #1 (Remarks to the Author):

I am totally satisfied with the revisions made by the authors, in which all my comments have been addressed. Now the reaction mechanism of MDD proposed by the authors is updated appropriately and agrees well with recent knowledges of the enzyme. The comparison of the structures of MDD solved in the study with the previously known structures of the enzyme and its orthologs, considering the effect of crystal packing, makes sense. I still hope the catalytic mechanism of this interesting decarboxylase is elucidated by QM/MM study, but it is not necessary in this paper. The elucidation will be made in future probably by the authors or other groups.

We appreciate your the comments and hope to address the QM/MM study in the future.

Reviewer #2 (Remarks to the Author):

In the manuscript entitled "Visualizing the Enzyme Mechanism of Mevalonate Diphosphate Decarboxylase" by Chen et al. (Manuscript NCOMMS-19-27996A) the authors submitted a revised version of their manuscript.

The revised version has substantially improved.

(1) The structural comparison of different MDD structures available in the PDB improves the understanding of the differences of the structures for example different crystal packing effects and hence restricted conformational flexibility of the elements. See also the novel Supplementary Figure 7. Here, I just have a minor point: I personally find it useful if the PDB-ID for the displayed structure (panel c to e) is provided and/or a reference to the primary citation. This suggestion applies as well to Supplementary figures.

We have added all the PDB codes to Supplementary Figures as requested.

The functional implications are now much better to follow. In particular, the discussion of the putative function of Asp282 is now better to understand. By the discussion of the other two recent publications, as suggested by the other reviewer, the authors were able to discard the contested hypothesis section on the mechanism. It is a very beautiful example how structural biology can provide insightful structural information to elucidate an enzymatic mechanism.

I recommend to accept the current version of the manuscript.

Thank you for the comments.

Minor points:

- Page 12, line 11: Names of organism should be in italic

We have edited the context in Page 12, line 16.

- Supplementary Figure 5: I suggest deleting the primer sequences. They are given in the Materials and Methods section.

We have removed the primer sequences in Supplementary Figure 5.